# Fixing Value Function Decomposition for Multi-Agent Reinforcement Learning

## Abstract

Value function decomposition methods for cooperative multi-agent reinforcement learning combine individual per-agent utilities into joint values trained on a joint objective. To ensure consistent action selection between individual utilities and joint values, it is imperative for the composition to satisfy *individual-global max* (IGM). However, most methods that satisfy IGM are characterized by limited representation capabilities that hinder their performance, and the one known exception is unnecessarily convoluted. In this work, we reveal a minimalistic formulation of IGM that inspires the derivation of QFIX, a novel family of value function decomposition methods that expand the representation capabilities of prior methods by means of a small "fixing" network. We implement three variants of QFIX, and demonstrate empirically that QFIX is able to meet or exceed state-of-the-art performance with better stability.

## 1. Introduction

Centralized training for decentralized execution (CTDE) (Lowe et al., 2017) is a powerful framework for cooperative multi-agent reinforcement learning (MARL) characterized by a centralized training phase where privileged information is freely shared between agents and a decentralized execution phase where agents act independently in adherence to standard decentralized control. As a consequence of a training phase that is informed by the full team's behavior and experiences (and, when feasible, the environment state), CTDE is commonly associated with increased coordination between agents and superior performances.

Value function decomposition (Sunehag et al., 2017) is a class of CTDE methods that construct a joint team value from individual per-agent utilities that encode agent be-

haviors. By training the joint value on a joint centralized objective, the individual utilities are also indirectly trained, resulting in decentralized agent policies that can be executed independently. Since its inception, value function decomposition has become a topic of great interest in cooperative MARL, with significant research effort put in both practical algorithms (Sunehag et al., 2017; Son et al., 2019; Rashid et al., 2020a;b; Wang et al., 2020; Marchesini et al., 2024) and theoretical understanding (Wang et al., 2021; Marchesini et al., 2024). *Individual-global max* (IGM) (Son et al., 2019) has been identified as a key property that connects individual utilities and joint values, ensuring that their associated decision making processes remain consistent.

In this work, we advance both theory and practice of value function decomposition. We formulate a novel *minimalistic* formulation of IGM-complete value function decomposition. Our formulation (i) correctly addresses general decentralized partially observable control (avoiding strong assumptions like full observability or centralized control), and (ii) highlights the core mechanism that characterizes the full IGM-complete function class. In contrast, prior methods fail to satisfy at least one of these criteria (usually the first). We introduce QFIX, a novel family of value function decomposition methods inspired by our formulation of IGM-complete decomposition. QFIX employs a simple "fixing" network to extend the representation capabilities of prior methods. We derive two main specializations of QFIX called QFIX-sum and QFIX-mono, respectively obtained by "fixing" VDN (Sunehag et al., 2017) and QMIX (Rashid et al., 2020b). To provide further insights into the core mechanisms that make value function decomposition so effective, we also derive QFIX-lin, a third variant that technically falls outside of the QFIX family, but combines QFIX-sum with a core component of QPLEX. Finally, we extend prior work on stateful value function decomposition to QFIX. An empirical evaluation on the StarCraft Multi-Agent Challenge v2 (Ellis et al., 2023) demonstrates that QFIX (i) is effective at enhancing prior non-IGM-complete methods like VDN and QMIX, (ii) is simpler to implement and understand, and require smaller models than QPLEX, a state-of-the-art method in IGM-complete value function decomposition, (iii) is competitive or outperforms QPLEX while also showing more stable convergence.

---

[1]Anonymous Institution, Anonymous City, Anonymous Region, Anonymous Country. Correspondence to: Anonymous Author <anon.email@domain.com>.

Preliminary work. Under review by the International Conference on Machine Learning (ICML). Do not distribute.

## 2. Related Work

Value Decomposition Networks (VDN) (Sunehag et al., 2017) are a precursor to value decomposition methods that employ a simple additive composition of individual utilities. QMIX (Rashid et al., 2020b) employs a monotonic composition that generalizes the function class of VDN resulting in significant performance improvements. Since VDN and QMIX have limited expressiveness, several models have attempted to achieve a broader function class. Weighted-QMIX (WQMIX) (Rashid et al., 2020a) aims to expand the function class of QMIX to non-monotonic cases so as to include optimal values $Q^*$. However, WQMIX appears to conflate the possibility of exploiting state information during centralized training (which is correct) with the goal of learning the decision process for a team of fully observable agents (which is incorrect). As a consequence, the theory of WQMIX assumes state values $\hat{Q}(s, \boldsymbol{a})$ and an optimization process that aims to recover the optimal fully observable decision making process $\operatorname{argmax}_{\boldsymbol{a}} Q^*(s, \boldsymbol{a})$, which is inconsistent with partially observable decentralized control. In contrast, QFIX is fully consistent with general partially observable decentralized control. Son et al. (2019) identify *individual-global max* (IGM) as an important property that corresponds to consistency between the individual and joint decision making processes. Notably, VDN and QMIX satisfy IGM, but are unable to represent the entire IGM-complete function class. QTRAN (Son et al., 2019) identifies a set of constraints that are sufficient to imply IGM, and employs auxiliary objectives that softly enforce those constraints. Son et al. (2019) argue that their constraints are also necessary for IGM under affine transformations, however they only show that one such affine transformation exists, rather than IGM being satisfied for all affine transformations. In contrast, QFIX is both sufficient and necessary to imply IGM, thus directly achieving the full IGM-complete function class. QPLEX (Wang et al., 2020) employs a dueling network decomposition and multiple layers of transformations to achieve the IGM-complete function class. However, QPLEX employs complex transformations that are superfluous in relation to its representation capabilities, and fails to identify the core underlying mechanism that is ultimately responsible to achieve the IGM function class. In contrast, QFIX is simpler to understand, and achieves the IGM function class with smaller models. Further, QPLEX is only one instance in the space of IGM-complete models, and our work will allow researchers to explore other instances that can further improve performance while adhering to IGM.

## 3. Background

### 3.1. Decentralized Multi-Agent Control

A decentralized POMDP (Dec-POMDP) (Oliehoek & Amato, 2016) generalizes single-agent partially ob-

servable control by accounting for multiple decentralized agents acting concurrently to solve a shared cooperative task. A Dec-POMDP is defined by a tuple $\langle N, \mathcal{S}, \{\mathcal{A}_1, \ldots, \mathcal{A}_N\}, \{\mathcal{O}_1, \ldots, \mathcal{O}_N\}, T, R, O, \gamma \rangle$ composed of: (i) number of agents $N \geq 2$; (ii) state space $\mathcal{S}$; (iii) individual action and observation spaces, respectively $\mathcal{A}_i$ and $\mathcal{O}_i$; (iv) starting state distribution $p \in \Delta \mathcal{S}$; (v) state transition function $T \colon \mathcal{S} \times \boldsymbol{\mathcal{A}} \to \Delta \mathcal{S}$; (vi) joint observation function $O \colon \boldsymbol{\mathcal{A}} \times \mathcal{S} \to \Delta \mathcal{O}$; (vii) joint reward function $R \colon \mathcal{S} \times \boldsymbol{\mathcal{A}} \to \mathbb{R}$; (viii) discount factor $\gamma \in [0, 1)$.

The number of agents $N$ determines a set of agent indices $\mathcal{I} \doteq \{1, \ldots, N\}$. The joint action, observation, and history spaces are defined as the respective Cartesian products $\boldsymbol{\mathcal{A}} \doteq \times_i \mathcal{A}_i$, $\boldsymbol{\mathcal{O}} \doteq \times_i \mathcal{O}_i$, and $\boldsymbol{\mathcal{H}} \doteq \times_i \mathcal{H}_i$. Therefore, joint actions $\boldsymbol{a} = (a_1, \ldots, a_N)$, observations $\boldsymbol{o} = (o_1, \ldots, o_N)$, and histories $\boldsymbol{h} = (h_1, \ldots, h_N)$ are tuples of the respective individual actions, observations, and histories.

Individual agent behaviors are generally modeled as individual stochastic policies $\pi_i \colon \mathcal{H}_i \to \Delta \mathcal{A}_i$ that act based on their respective history $h_i \in \mathcal{H}_i \doteq \mathcal{O}_i \times (\mathcal{A}_i \times \mathcal{O}_i)^*$. The combined behavior of all policies is represented as a joint (but still decentralized) policy $\boldsymbol{\pi}(\boldsymbol{h}, \boldsymbol{a}) \doteq \prod_i \pi_i(h_i, a_i)$ that factorizes accordingly. Decentralized multi-agent control aims to find policies that jointly maximize the expected sum of discounted rewards $J^{\boldsymbol{\pi}} \doteq \mathbb{E}\left[\sum_t \gamma^t R(s_t, \boldsymbol{a}_t)\right]$.

In this work, we focus on approaches that model agent policies implicitly via parametric utilities $\hat{Q}_i \colon \mathcal{H}_i \times \mathcal{A}_i \to \mathbb{R}$, typically by means of greedy or $\epsilon$-greedy action selection. Such utilities $\hat{Q}_i(h_i, a_i)$ are commonly decomposed into corresponding values $\hat{V}_i(h_i) \doteq \max_{a_i} \hat{Q}_i(h_i, a_i)$ and (non-positive) advantages $\hat{A}_i(h_i, a_i) \doteq \hat{Q}_i(h_i, a_i) - \hat{V}_i(h_i)$. When convenient, we occasionally employ shorthand notation $q_i \doteq \hat{Q}_i(h_i, a_i)$, $v_i \doteq \hat{V}_i(h_i)$, and $u_i \doteq \hat{A}_i(h_i, a_i)$.

### 3.2. Value Function Decomposition

Value function decomposition methods (Sunehag et al., 2017; Rashid et al., 2020b; Wang et al., 2020) construct joint values $\hat{Q}(\boldsymbol{h}, \boldsymbol{a})$ from individual per-agent *utilities* $\hat{Q}_i(h_i, a_i)$. We specifically use the term *utility* here to underscore the fact that $\hat{Q}_i(h_i, \cdot)$ represents an ordering over actions, rather than any notion of expected performance. Notably, $\hat{Q}_i$ is never trained to perform evaluation, and neither $\hat{Q}_i(h_i, a_i) \approx Q_i^{\pi}(h_i, a_i)$ nor $\hat{Q}_i(h_i, a_i) \approx Q_i^*(h_i, a_i)$ are expected interpretations of well-trained utilities $\hat{Q}_i$.

Value function decomposition methods employ joint models $\hat{Q}(\boldsymbol{h}, \boldsymbol{a})$ that are a function of the individual utilities $\hat{Q}(h_i, a_i)$, and mainly differ in terms of the relationship that is enforced and the corresponding emergent properties. The joint model $\hat{Q}(\boldsymbol{h}, \boldsymbol{a})$ is trained on a *joint* objective,

$$\mathcal{L}_{\hat{Q}}(\boldsymbol{h}, \boldsymbol{a}, r, \boldsymbol{o}) \doteq \frac{1}{2}\left(r + \gamma \max_{\boldsymbol{a}'} \hat{Q}^-(\boldsymbol{h}\boldsymbol{a}\boldsymbol{o}, \boldsymbol{a}') - \hat{Q}(\boldsymbol{h}, \boldsymbol{a})\right)^2,$$
$$(1)$$

which indirectly trains the individual utilities and behaviors.

### 3.2.1. INDIVIDUAL-GLOBAL MAX

Son et al. (2019) identify individual-global max (IGM) as a useful property of decomposition models to achieve decentralized action selection and address scaling concerns.

**Definition 3.1** (Individual-Global Max). Individual utilities $\{Q_i(h_i, a_i)\}_{i=1}^N$ and joint values $Q(\boldsymbol{h}, \boldsymbol{a})$ satisfy *individual-global max* (IGM) iff

$$\operatorname*{argmax}_{a_i} Q_i(h_i, a_i) = \left( \operatorname*{argmax}_{\boldsymbol{a}} Q(\boldsymbol{h}, \boldsymbol{a}) \right)_i. \quad (2)$$

IGM denotes whether the individual and global decision making processes are equivalent, and reduces the complexity of finding the maximal joint action from exponential to linear in the number of agents: For a given joint history $\boldsymbol{h}$, the full search over the joint action space $\boldsymbol{\mathcal{A}}$ can be replaced with $N$ independent searches over the individual action spaces $\mathcal{A}_i$. VDN (Section 3.2.2) and QMIX (Section 3.2.3) are well-known models that satisfy IGM, although their function class are limited subsets of all IGM values.

### 3.2.2. VDN: ADDITIVE DECOMPOSITION

Value Decomposition Networks (VDN) (Sunehag et al., 2017) is a precursor to value function decomposition methods. VDN employs a simple additive value decomposition,

$$\hat{Q}_{\text{VDN}}(\boldsymbol{h}, \boldsymbol{a}) \doteq \sum_i \hat{Q}_i(h_i, a_i). \quad (3)$$

### 3.2.3. QMIX: MONOTONIC DECOMPOSITION

QMIX (Rashid et al., 2020b) constructs joint values as a *monotonic* function of individual utilities,

$$\hat{Q}_{\text{MIX}}(\boldsymbol{h}, \boldsymbol{a}) \doteq f_{\text{mono}}(q_1, \dots, q_N), \quad (4)$$

where $f_{\text{mono}} \colon \mathbb{R}^N \to \mathbb{R}$ is a parametric mixing network that satisfies monotonicity,

$$\frac{\partial f_{\text{mono}}(q_1, \dots, q_N)}{\partial q_i} \geq 0. \quad (5)$$

Monotonic composition generalizes the additive composition of VDN, consequently achieving a broader function class, though it still falls short from modeling the entire IGM function class. As in VDN, the joint model $\hat{Q}_{\text{MIX}}(\boldsymbol{h}, \boldsymbol{a})$ is trained on the centralized objective in Equation (1).

### 3.2.4. QPLEX: IGM-COMPLETE DECOMPOSITION

QPLEX (Wang et al., 2020) reframes IGM in terms of advantages, and employs dueling network decomposition to achieve full function class equivalence with IGM.

**Definition 3.2** (IGM-Complete Function Class). A function class of individual utilities $\{Q_i(h_i, a_i)\}_{i=1}^N$ and joint values $Q(\boldsymbol{h}, \boldsymbol{a})$ is IGM-complete if it contains all and only functions that satisfy IGM.

Given utilities $Q_i(h_i, a_i)$ and joint action-values $Q(\boldsymbol{h}, \boldsymbol{a})$, corresponding values and advantages are defined as follows,

$$V_i(h_i) \doteq \max_{a_i} Q_i(h_i, a_i), \quad A_i(h_i, a_i) \doteq Q_i(h_i, a_i) - V_i(h_i), \quad (6)$$

$$V(\boldsymbol{h}) \doteq \max_{\boldsymbol{a}} Q(\boldsymbol{h}, \boldsymbol{a}), \qquad A(\boldsymbol{h}, \boldsymbol{a}) \doteq Q(\boldsymbol{h}, \boldsymbol{a}) - V(\boldsymbol{h}). \quad (7)$$

Wang et al. (2020) reformulate IGM as a set of numeric constraints between these individual and joint advantages.

**Definition 3.3** (Advantage Constraints). Individual utilities $\{Q_i(h_i, a_i)\}_{i=1}^N$ and joint values $Q(\boldsymbol{h}, \boldsymbol{a})$ satisfy IGM iff, $\forall \boldsymbol{h} \in \boldsymbol{\mathcal{H}}, \forall \boldsymbol{a}^* \in \boldsymbol{\mathcal{A}}^*(\boldsymbol{h})$, and $\forall \boldsymbol{a} \in \boldsymbol{\mathcal{A}} \setminus \boldsymbol{\mathcal{A}}^*(\boldsymbol{h})$,

$$A(\boldsymbol{h}, \boldsymbol{a}^*) = 0, \qquad A_i(h_i, a_i^*) = 0, \quad (8)$$
$$A(\boldsymbol{h}, \boldsymbol{a}) < 0, \qquad A_i(h_i, a_i) \leq 0, \quad (9)$$

where $\boldsymbol{\mathcal{A}}^*(\boldsymbol{h}) \doteq \{\boldsymbol{a} \in \boldsymbol{\mathcal{A}} \mid Q(\boldsymbol{h}, \boldsymbol{a}) = V(\boldsymbol{h})\}$ is the subset of maximal joint actions according to the joint values.

QPLEX employs a mixing structure that provably enforces Definition 3.3. Individual utilities $\hat{Q}_i(h_i, a_i)$ are first decomposed into $\hat{V}_i(h_i)$ and $\hat{A}_i(h_i, a_i)$, and then transformed using centralized joint history information as follows,

$$\hat{V}_i(\boldsymbol{h}) \doteq w_i(\boldsymbol{h})\hat{V}_i(h_i) + b_i(\boldsymbol{h}), \quad (10)$$
$$\hat{A}_i(\boldsymbol{h}, a_i) \doteq w_i(\boldsymbol{h})\hat{A}_i(h_i, a_i), \quad (11)$$

where $w_i \colon \boldsymbol{\mathcal{H}} \to \mathbb{R}_{>0}$ are parametric positive weights and $b_i \colon \boldsymbol{\mathcal{H}} \to \mathbb{R}$ are parametric biases. These transformed values are aggregated as weighted sums,

$$\hat{V}_{\text{PLEX}}(\boldsymbol{h}) \doteq \sum_i \hat{V}_i(\boldsymbol{h}), \quad (12)$$
$$\hat{A}_{\text{PLEX}}(\boldsymbol{h}, \boldsymbol{a}) \doteq \sum_i \lambda_i(\boldsymbol{h}, \boldsymbol{a})\hat{A}_i(\boldsymbol{h}, a_i), \quad (13)$$

where $\lambda_i \colon \boldsymbol{\mathcal{H}} \times \boldsymbol{\mathcal{A}} \to \mathbb{R}_{>0}$ are parametric positive weights. Finally, $\hat{Q}_{\text{PLEX}}(\boldsymbol{h}, \boldsymbol{a})$ is obtained by recombining aggregate values and advantages,

$$\hat{Q}_{\text{PLEX}}(\boldsymbol{h}, \boldsymbol{a}) \doteq \hat{V}_{\text{PLEX}}(\boldsymbol{h}) + \hat{A}_{\text{PLEX}}(\boldsymbol{h}, \boldsymbol{a}). \quad (14)$$

This sequence of decomposition, transformations, and recomposition, combined with positive weights $w_i$ and $\lambda_i$ results in the constraint from Definition 3.3 being satisfied. Wang et al. (2020) also demonstrate that QPLEX satisfies Definition 3.2 and its function class is IGM-complete, given sufficiently expressive models $w_i(\boldsymbol{h})$, $b_i(\boldsymbol{h})$, and $\lambda_i(\boldsymbol{h}, \boldsymbol{a})$.

### 3.2.5. STATEFUL VALUE FUNCTION DECOMPOSITION

Practical implementations of value function decomposition methods often employ stateful joint values $Q(\boldsymbol{h}, s, \boldsymbol{a})$ and

diverge from the stateless theoretical derivations in ways that may undermine core IGM-related properties. To address the effects of state in value function decomposition, Marchesini et al. (2024) formulate a state-compliant version of IGM.

**Definition 3.4** (Stateful-IGM). Utilities $\{Q_i(h_i, a_i)\}_{i=1}^N$ and stateful joint values $Q(\boldsymbol{h}, s, \boldsymbol{a})$ satisfy IGM iff

$$\arg\max_{a_i} Q_i(h_i, a_i) = \left(\arg\max_{\boldsymbol{a}} \mathbb{E}_{s|\boldsymbol{h}}\left[Q(\boldsymbol{h}, s, \boldsymbol{a})\right]\right)_i \quad (15)$$

Marchesini et al. (2024) show that the stateful implementations of QMIX and QPLEX continue to satisfy IGM, while the stateful implementation of QPLEX (which employs historyless stateful weights $w_i(s), \lambda_i(s, \boldsymbol{a})$) is not IGM-complete. Nonetheless, stateful implementations often perform well in practice, and remain a common occurrence.

## 4. Fixing Value Function Decomposition

Although QPLEX achieves the IGM-complete function class, it is expressed as a convoluted sequence of transformations that are never fully motivated. Unrolling the QPLEX values directly in terms of individual values, we get

$$\hat{Q}_{\text{PLEX}}(\boldsymbol{h}, \boldsymbol{a}) = \sum_i w_i(\boldsymbol{h})\hat{V}_i(h_i) + b_i(\boldsymbol{h}) \\ + w_i(\boldsymbol{h})\lambda_i(\boldsymbol{h}, \boldsymbol{a})\hat{A}_i(h_i, a_i), \quad (16)$$

which raises questions about which components of this structure are truly important or necessary, e.g., the product of individual advantages with two types of positive weights $w_i(\boldsymbol{h})$ and $\lambda_i(\boldsymbol{h}, \boldsymbol{a})$ appears to be redundant. QPLEX only represents one instance in a space of models that achieve IGM-completeness, and whether simpler better-performing decompositions exist remains an open question. The convoluted nature of the QPLEX transformations motivate us to find a simpler and more general formulation of IGM-complete decomposition.

In this section, we first propose a minimal formulation of IGM-complete value function decomposition. Then, we use this formulation to develop *QFIX*, a novel family of value function decomposition models that operate by expanding the representation capabilities of prior non-IGM-complete models. We derive two primary instances of QFIX based on "fixing" VDN and QMIX respectively, and a third instance designed to resemble QPLEX. We also derive *additive QFIX* (Q+FIX), a simple variant of QFIX that achieves significant practical performance gains, and derive Q+FIX counterparts of the QFIX instances. Finally, we discuss stateful variants of QFIX and how state affects its theoretical properties.

### 4.1. A Minimal Formulation of IGM-Complete Values

We aim to formalize IGM-complete value function decomposition in its simplest and most essential form. We begin

by simplifying Definition 3.3, noting that three of the four constraints are satisfied by definition; The only constraint that requires active enforcement is $A_i(h_i, a_i^*) = 0$.

**Definition 4.1** (Simplified Advantage Constraints). Utilities $\{Q_i(h_i, a_i)\}_{i=1}^N$ and joint values $Q(\boldsymbol{h}, \boldsymbol{a})$ satisfy IGM iff,

$$A(\boldsymbol{h}, \boldsymbol{a}) = 0 \implies \forall i\left(A_i(h_i, a_i) = 0\right), \quad (17)$$

or, equivalently via contraposition,

$$\exists i\left(A_i(h_i, a_i) \neq 0\right) \implies A(\boldsymbol{h}, \boldsymbol{a}) \neq 0. \quad (18)$$

In essence, constructing joint advantages $A(\boldsymbol{h}, \boldsymbol{a})$ that are negative iff any of the individual advantages $A_i(h_i, a_i)$ are negative is both sufficient and necessary to satisfy IGM.

Consider the aptly named function

$$Q_{\text{IGM}}(\boldsymbol{h}, \boldsymbol{a}) \doteq w(\boldsymbol{h}, \boldsymbol{a})f(u_1, \ldots, u_N) + b(\boldsymbol{h}), \quad (19)$$

where $u_i = A_i(h_i, a_i)$ are the individual advantages, $w: \mathcal{H} \times \mathcal{A} \to \mathbb{R}_{>0}$ is an arbitrary positive function of joint history and joint action, $b: \mathcal{H} \to \mathbb{R}$ is an arbitrary function of joint history, and $f: \mathbb{R}_{\leq 0}^n \to \mathbb{R}_{\leq 0}$ is any non-positive function that is zero iff all inputs are zero (e.g., $f(u_1, \ldots, u_N) = \sum_i u_i$ is a simple instance of $f$). We note

$$V_{\text{IGM}}(\boldsymbol{h}) \doteq \max_{\boldsymbol{a}} Q_{\text{IGM}}(\boldsymbol{h}, \boldsymbol{a}) \\ = b(\boldsymbol{h}), \quad (20)$$
$$A_{\text{IGM}}(\boldsymbol{h}, \boldsymbol{a}) \doteq Q_{\text{IGM}}(\boldsymbol{h}, \boldsymbol{a}) - V_{\text{IGM}}(\boldsymbol{h}) \\ = w(\boldsymbol{h}, \boldsymbol{a})f(u_1, \ldots, u_N). \quad (21)$$

Essentially, $Q_{\text{IGM}}$ denotes a relationship where any deviation from individual maximality (characterized by at least one negative utility $u_i < 0$, and corresponding to a negative $f(u_1, \ldots, u_N) < 0$) is transformed into an arbitrary deviation $w(\boldsymbol{h}, \boldsymbol{a})f(u_1, \ldots, u_N) < 0$ from joint maximality. Per Definition 4.1, $Q_{\text{IGM}}$ represents the IGM function class.

**Lemma 4.2.** *For any $f$, $w$, and $b$, values $\{Q_i\}_{i=1}^N$ and $Q_{\text{IGM}}$ satisfy IGM. (See proof in Appendix A.1.)*

**Theorem 4.3.** *For any $f$, and given free choice of $w$ and $b$, the function class of $\{Q_i\}_{i=1}^N$ and $Q_{\text{IGM}}$ is IGM-complete. (See proof in Appendix A.2.)*

$Q_{\text{IGM}}$ is a minimal formulation of the IGM function class based on a single weighted transformation of individual advantages. Next, we explore how this formulation can be used to derive QFIX, a novel family of value function decomposition models that work by expanding the representation capabilities of prior non-IGM-complete models.

### 4.2. QFIX

Let $\hat{Q}_{\text{fixee}}(\boldsymbol{h}, \boldsymbol{a})$ denote a "fixee" value function decomposition model that satisfies IGM but is not IGM-complete, e.g.,

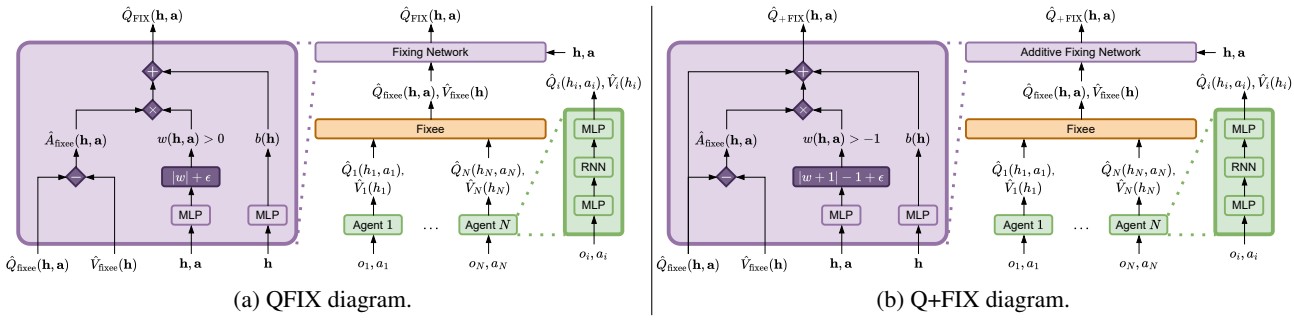

(a) QFIX diagram.       (b) Q+FIX diagram.

*Figure 1.* QFIX and Q+FIX diagrams.

VDN or QMIX. Equation (19) suggests a method to "fix" $\hat{Q}_{\text{fixee}}$ and have it achieve full IGM-completeness. We can extend the expressiveness of $\hat{Q}_{\text{fixee}}$ by processing it through a "fixing" network that resembles Equation (19),

$$\hat{Q}_{\text{FIX}}(\boldsymbol{h}, \boldsymbol{a}) \doteq w(\boldsymbol{h}, \boldsymbol{a})\hat{A}_{\text{fixee}}(\boldsymbol{h}, \boldsymbol{a}) + b(\boldsymbol{h}), \quad (22)$$

where $w\colon \mathcal{H} \times \mathcal{A} \to \mathbb{R}_{>0}$ is a parametric positive model, $b\colon \mathcal{H} \to \mathbb{R}$ is a parametric model, and $\hat{A}_{\text{fixee}}\colon \mathcal{H} \times \mathcal{A} \to \mathbb{R}_{\leq 0}$ is the non-positive advantage of the fixee as defined by

$$\hat{V}_{\text{fixee}}(\boldsymbol{h}) \doteq \max_{\boldsymbol{a}} \hat{Q}_{\text{fixee}}(\boldsymbol{h}, \boldsymbol{a}), \quad (23)$$

$$\hat{A}_{\text{fixee}}(\boldsymbol{h}, \boldsymbol{a}) \doteq \hat{Q}_{\text{fixee}}(\boldsymbol{h}, \boldsymbol{a}) - \hat{V}_{\text{fixee}}(\boldsymbol{h}). \quad (24)$$

Figure 1a shows a diagram of the QFIX fixing structure. We note that $\hat{A}_{\text{fixee}}(\boldsymbol{h}, \boldsymbol{a})$ is zero iff the joint action $\boldsymbol{a}$ is maximal according to $\hat{Q}_{\text{fixee}}$, and negative otherwise. Given that $\hat{Q}_{\text{fixee}}$ satisfies IGM by assumption, $\boldsymbol{a}$ is maximal according to $\hat{Q}_{\text{fixee}}$ iff the individual actions $a_i$ are maximal according to $\hat{Q}_i(h_i, a_i)$, or, equivalently, iff $\hat{A}_i(h_i, a_i) = 0$. In short, $\hat{A}_{\text{fixee}}(\boldsymbol{h}, \boldsymbol{a})$ satisfies the requirements of $f$ in Equation (19).

**Theorem 4.4.** *QFIX satisfies IGM. Given sufficiently expressive $w$ and $b$, the function class of QFIX is IGM-complete. (See proof in Appendix A.3.)*

Given the free choice of fixee model $\hat{Q}_{\text{fixee}}$, QFIX really represents a family of value function decomposition models. This enables us to consider more or less complex fixees (e.g., VDN vs QMIX) to find an acceptable tradeoff between minimizing the complexity of the fixee model, and minimizing the "fixing" burden on the fixing network.

Next, we compare QFIX to QPLEX, present two primary instances of QFIX based on fixing VDN and QMIX, and present yet another variant inspired by QPLEX.

### 4.2.1. RELATIONSHIP TO QPLEX

The advantage component of QFIX, $w(\boldsymbol{h}, \boldsymbol{a})\hat{A}_{\text{fixee}}(\boldsymbol{h}, \boldsymbol{a})$, is similar to one of the transformations of QPLEX, $\sum_i \lambda_i(\boldsymbol{h}, \boldsymbol{a})\hat{A}_i(\boldsymbol{h}, a_i)$ (see Equation (13)), which also applies positive weights to transformed aggregates of the individual advantages. This similarity is no coincidence, as it

is specifically that component of QPLEX that is singularly responsible for ensuring IGM-completeness; it is a more convoluted form of our proposed fixing structure. However, QPLEX also employs various other transformations that do not contribute to achieving the IGM-complete function class, and their necessity remains questionable (beyond general considerations of modelling structure and size).

The weights $\lambda_i(\boldsymbol{h}, \boldsymbol{a})$ employed by QPLEX are also more complex in that there is one such model per agent, and each is implemented via self-importance. In contrast, we employ a simpler structure based on a single model implemented as a feed-forward network, and still manage to achieve performance improvements. Our formulation is simpler in that it focuses entirely on this single transformation, which is minimally sufficient to guarantee IGM-completeness.

### 4.2.2. QFIX-SUM: FIXING VDN.

QFIX-sum is an instance of QFIX based on VDN, i.e., with $\hat{Q}_{\text{fixee}}(\boldsymbol{h}, \boldsymbol{a}) = \hat{Q}_{\text{VDN}}(\boldsymbol{h}, \boldsymbol{a})$, which results in (see Appendix B.3 for the formal derivation)

$$\hat{Q}_{\text{FIX-sum}}(\boldsymbol{h}, \boldsymbol{a}) = w(\boldsymbol{h}, \boldsymbol{a}) \sum_i \hat{A}_i(h_i, a_i) + b(\boldsymbol{h}). \quad (25)$$

### 4.2.3. QFIX-MONO: FIXING QMIX

QFIX-mono is an instance of QFIX based on QMIX, i.e., with $\hat{Q}_{\text{fixee}}(\boldsymbol{h}, \boldsymbol{a}) = \hat{Q}_{\text{MIX}}(\boldsymbol{h}, \boldsymbol{a})$, which results in (see Appendix B.4 for formal derivation)

$$\begin{aligned}\hat{Q}_{\text{FIX-mono}}(\boldsymbol{h}, \boldsymbol{a}) = w(\boldsymbol{h}, \boldsymbol{a}) \, (&f_{\text{mono}}(q_1, \dots, q_N) \\ &- f_{\text{mono}}(v_1, \dots, v_N)) \\ + b(\boldsymbol{h}). \quad &(26)\end{aligned}$$

### 4.2.4. QFIX-LIN: SIMPLIFYING QPLEX

Given the similarity between QFIX and QPLEX shown in Section 4.2.1, we may consider yet another QFIX variant that also applies per-agent positive weights $w_i(\boldsymbol{h}, \boldsymbol{a}) > 0$, similarly to QPLEX. Due to the linear structure that strictly generalizes the sum of QFIX-sum (though both achieve

IGM-completeness), we may call this variant QFIX-lin.

$$\hat{Q}_{\text{FIX-lin}}(\boldsymbol{h}, \boldsymbol{a}) \doteq \sum_i w_i(\boldsymbol{h}, \boldsymbol{a}) \hat{A}_i(h_i, a_i) + b(\boldsymbol{h}) . \quad (27)$$

QFIX-lin does not strictly satisfy the form of Equation (22), however, it represents a close enough variant of QFIX-sum that we consider it QFIX-adjacent and name it accordingly. QFIX-lin is a strict generalization of QFIX-sum, which can be recovered as a special case where all the weights $w_i(\boldsymbol{h}, \boldsymbol{a})$ are equal. Formally, we must still explicitly prove the IGM properties of QFIX-lin.

**Theorem 4.5.** *QFIX-lin satisfies IGM. Given sufficiently expressive $w_i$ and $b$, the function class of QFIX-lin is IGM-complete. (See proof in Appendix A.4.)*

### 4.2.5. RECOVERING THE FIXEE MODEL

We take a moment to note that QFIX is able to recover the fixee model via $w(\boldsymbol{h}, \boldsymbol{a}) = 1$ and $b(\boldsymbol{h}) = \hat{V}_{\text{fixee}}(\boldsymbol{h})$,

$$\begin{aligned} \hat{Q}_{\text{FIX}}(\boldsymbol{h}, \boldsymbol{a}) &= w(\boldsymbol{h}, \boldsymbol{a}) \hat{A}_{\text{fixee}}(\boldsymbol{h}, \boldsymbol{a}) + b(\boldsymbol{h}) \\ &= 1 \cdot \hat{A}_{\text{fixee}}(\boldsymbol{h}, \boldsymbol{a}) + \hat{V}_{\text{fixee}}(\boldsymbol{h}) \\ &= \hat{Q}_{\text{fixee}}(\boldsymbol{h}, \boldsymbol{a}) . \end{aligned} \quad (28)$$

Such values of $w(\boldsymbol{h}, \boldsymbol{a})$ and $b(\boldsymbol{h})$ establish a direct relationship between the fixee and fixed models, which is relevant as we next use this relationship to derive a theoretically equivalent but better-performing *additive* variant of QFIX.

## 4.3. Additive QFIX (Q+FIX)

In this section, we further derive a simple variant of QFIX which, albeit having the same theoretical properties, achieves significant practical performance improvements. This variant will take on an additive form, when compared to the fixee model, hence its name *additive QFIX* (Q+FIX).

As noted in Section 4.2.5, the values of $w(\boldsymbol{h}, \boldsymbol{a}) = 1$ and $b(\boldsymbol{h}) = \hat{V}_{\text{fixee}}(\boldsymbol{h})$ hold a special significance for QFIX. Q+FIX is derived by reparameterizing $w$ and $b$ to incorporate such values via simple addition, as follows,

$$\begin{aligned} &\hat{Q}_{\text{+FIX}}(\boldsymbol{h}, \boldsymbol{a}) \\ &\doteq (w(\boldsymbol{h}, \boldsymbol{a}) + 1) \hat{A}_{\text{fixee}}(\boldsymbol{h}, \boldsymbol{a}) + (b(\boldsymbol{h}) + \hat{V}_{\text{fixee}}(\boldsymbol{h})) \\ &= \hat{Q}_{\text{fixee}}(\boldsymbol{h}, \boldsymbol{a}) + w(\boldsymbol{h}, \boldsymbol{a}) \hat{A}_{\text{fixee}}(\boldsymbol{h}, \boldsymbol{a}) + b(\boldsymbol{h}) , \quad (29) \end{aligned}$$

where $w \colon \mathcal{H} \times \mathcal{A} \to \mathbb{R}_{>-1}$ is a parametric model constrained by $w(\boldsymbol{h}, \boldsymbol{a}) > -1$, $b \colon \mathcal{H} \to \mathbb{R}$ is a parametric model, and $\hat{Q}_{\text{fixee}}$ and $\hat{A}_{\text{fixee}}$ are the fixee action-values and advantages. Figure 1b shows a diagram of the Q+FIX fixing structure. This reparameterization allows Q+FIX to more directly exploit the original fixee model, providing the IGM-complete function class as a separate additive component. Note that, following the reparameterization of the $w$ model,

the constraint imposed on its output has changed: since it's the full addition $w(\boldsymbol{h}, \boldsymbol{a}) + 1 > 0$ that must satisfy the positivity constraint from QFIX, the corresponding constraint for Q+FIX is now $w(\boldsymbol{h}, \boldsymbol{a}) > -1$.

**Theorem 4.6.** *Q+FIX satisfies IGM. Given sufficiently expressive $w$ and $b$, the function class of Q+FIX is IGM-complete. (See proof in Appendix A.5.)*

### 4.3.1. Q+FIX-SUM, Q+FIX-MONO, AND Q+FIX-LIN

Here, we show the Q+FIX counterparts to QFIX-sum, QFIX-mono, and QFIX-lin, respectively called Q+FIX-sum, Q+FIX-mono, and Q+FIX-lin. See Appendices B.5 to B.7 for their corresponding derivations and graphical diagrams.

$$\begin{aligned} \hat{Q}_{\text{+FIX-sum}}(\boldsymbol{h}, \boldsymbol{a}) &= \sum_i \hat{Q}_i(h_i, a_i) \\ &+ w(\boldsymbol{h}, \boldsymbol{a}) \sum_i \hat{A}_i(h_i, a_i) + b(\boldsymbol{h}). \quad (30) \end{aligned}$$

$$\begin{aligned} \hat{Q}_{\text{+FIX-mono}}(\boldsymbol{h}, \boldsymbol{a}) &= f_{\text{mono}}(q_1, \dots, q_N) \\ &+ w(\boldsymbol{h}, \boldsymbol{a}) \left( f_{\text{mono}}(q_1, \dots, q_N) \right. \\ &\left. - f_{\text{mono}}(v_1, \dots, v_N) \right) \\ &+ b(\boldsymbol{h}) . \quad (31) \end{aligned}$$

$$\begin{aligned} \hat{Q}_{\text{+FIX-lin}}(\boldsymbol{h}, \boldsymbol{a}) &= \sum_i \hat{Q}_i(h_i, a_i) \\ &+ \sum_i w_i(\boldsymbol{h}, \boldsymbol{a}) \hat{A}_i(h_i, a_i) + b(\boldsymbol{h}). \quad (32) \end{aligned}$$

### 4.3.2. DETACHING THE ADVANTAGES

The additive form of Q+FIX enables the use of an implementation detail already employed by QPLEX that appears to significantly improve performance, i.e., the detachment of the advantages when computing gradients. This can be expressed using the stop-gradient operator[1] stop as follows,

$$\begin{aligned} \hat{Q}_{\text{+FIX}}(\boldsymbol{h}, \boldsymbol{a}) &= \hat{Q}_{\text{fixee}}(\boldsymbol{h}, \boldsymbol{a}) \\ &+ w(\boldsymbol{h}, \boldsymbol{a}) \operatorname{stop} \left[ \hat{A}_{\text{fixee}}(\boldsymbol{h}, \boldsymbol{a}) \right] + b(\boldsymbol{h}) \quad (33) \end{aligned}$$

The reason why detaching the advantages improves performance is not fully understood. Wang et al. (2020, Appendix B.2) argue that it (cit.) *"increases the optimization stability of the max operator of the dueling structure"*, in reference to dueling networks (Wang et al., 2016). However, the connection between the detach and dueling networks remains unclear. Instead, we hypothesize that detaching the advantage may mitigate adverse effects that the fixing

---

[1]The stop-gradient function is a mathematical anomaly whose value behaves like the identity function, $\operatorname{stop}[x] = x$, while its gradient behaves like the zero function, $\nabla_x \operatorname{stop}[x] = 0$. It is a functionality commonly provided by deep learning frameworks, e.g., pytorch provides this via the `Tensor.detach()` method.

*Table 1.* Mixer sizes for `Protoss` in number of parameters.

| Protoss | 5vs5 | 10vs10 | 20vs20 |
|---|---|---|---|
| QMIX | 38 k | 83 k | 201 k |
| QPLEX | 135 k | 326 k | 882 k |
| Q+FIX-sum | 20 k | 50 k | 138 k |
| Q+FIX-mono | 54 k | 180 k | 743 k |
| Q+FIX-lin | 21 k | 51 k | 140 k |

structure may have on the gradients $\nabla_{\theta_i}\hat{Q}_{+\text{FIX}}(\boldsymbol{h}, \boldsymbol{a})$ of the joint values w.r.t. the agent parameters $\theta_i$ (see Appendix C).

### 4.4. Stateful Variants

As with QMIX and QPLEX, we may consider stateful variants of QFIX that partially deviate from the stateless theory developed so far. Such variants warrant a discussion on the implications of employing state information on the corresponding theoretical properties (Marchesini et al., 2024). Different versions of stateful QFIX are possible by combining stateless/stateful fixees with stateless/stateful fixing networks. We briefly summarize the conclusions for two main stateful variants. See additional discussion in Appendix D.

**History-State QFIX**   When employing history-state fixing models $w(\boldsymbol{h}, s, \boldsymbol{a})$ and $b(\boldsymbol{h}, s)$, QFIX continues to both satisfy IGM and achieve the IGM-complete function class.

**State-Only QFIX**   When employing state-only fixing models $w(s, \boldsymbol{a})$ and $b(s)$, QFIX continues to satisfy IGM, but fails to achieve the IGM-complete function class.

As Q+FIX is a reparameterization of QFIX, its properties remain the same in this regard. These conclusions are comparable to those for stateful QPLEX (Marchesini et al., 2024).

## 5. Evaluation

We perform an empirical evaluation comparing Q+FIX-sum, Q+FIX-mono, and Q+FIX-lin to competitive baselines in the `pymarl2` (Ellis et al., 2023) multi-agent framework.

**StarCraft Multi-Agent Challenge**   `Pymarl2` provides baseline implementations for the StarCraft Multi-Agent Challenge v2 (SMACv2) (Ellis et al., 2023), a popular benchmark for cooperative multi-agent control based on the real-time strategy game StarCraft II. SMACv2 features two battling teams composed by configurable races, race-dependent and stochastically determined unit types, and team sizes. Our empirical evaluation is based on 9 common scenarios obtained by combining the 3 races (`Protoss`, `Terran`, and `Zerg`) with 3 team sizes (5vs5, 10vs10, and 20vs20). `Pymarl2` provides implementations for VDN, QMIX, and QPLEX, our available baselines.

**Implementation Details**   We note that `pymarl2` provides *stateful* implementations of QMIX and QPLEX. For QPLEX in particular, this means that state-only weights $w_i(s)$ and $\lambda_i(s, \boldsymbol{a})$ are employed. To maintain a fair comparison, our implementation of Q+FIX methods employs an analogous stateful implementation with state-only weights $w(s, \boldsymbol{a})$ for Q+FIX-sum and Q+FIX-mono, and $w_i(s, \boldsymbol{a})$ for Q+FIX-lin. QPLEX and Q+FIX implementations both employ gradient detaching as described in Section 4.3.2.

**Metrics**   SMACv2 logs various metrics pertaining to team performance, including the mean return and the mean winrate obtained as the ratio of episodes where the agents succeed in defeating the enemies. Although the winrate is a common metric used in prior work (e.g., Wang et al. (2020) use the winrate in their SMACv1 evaluation), we have found that winrates induce a different ordering over performances, i.e., it is possible to obtain a higher winrate while achieving a lower return, and vice versa. This indicates that the rewards of SMACv2 do not perfectly encode the task of defeating the enemies—a matter of reward design that is beyond the scope of this work. Since returns are the metric that the methods are directly trained to maximize, we prioritize returns as our primary evaluation metric. Appendix E contains additional results and discussion based on winrates.

**Results**   We execute 3 independent runs per model per scenario, and show learning performance for each in Figure 2. To exploit the total sum of collected data, we also show (normalized) aggregate returns across scenarios in Figure 3.

As expected, VDN fails to be a competitive baseline on its own for most scenarios, likely due to the well-known limited representation. Fixing VDN via Q+FIX-sum, we are able to overcome this limitation (as noted by the performance gap between VDN and Q+FIX-sum), expanding its representation space and reaching SOTA performance.

QMIX sometimes exhibits fast initial learning speeds, albeit often to a sub-competitive final performance (`Protoss-5vs5`, `Terran-5vs5`, `Terran-10vs10`, `Zerg-10vs10`, `Terran-20vs20`, `Zerg-20vs20`), again a likely consequence of its limited representation. Fixing QMIX via Q+FIX-mono, we are often able to exploit the initial learning speeds and complement them with improved performance at convergence reaching SOTA performance.

QPLEX is highly competitive and performs very well in some scenarios (`Protoss-5vs5`, `Protoss-20vs20`, `Terran-20vs20`, `Zerg-20vs20`), but underperforms in others (`Terran-5vs5`, `Protoss-10vs10`, `Zerg-10vs10`), and exhibits troubling convergence instabilities as well (`Zerg-5vs5`, `Terran-10vs10`). Q+FIX-lin, as the simplified variant inspired by QPLEX, manages to avoid such convergence instabilities, plausibly

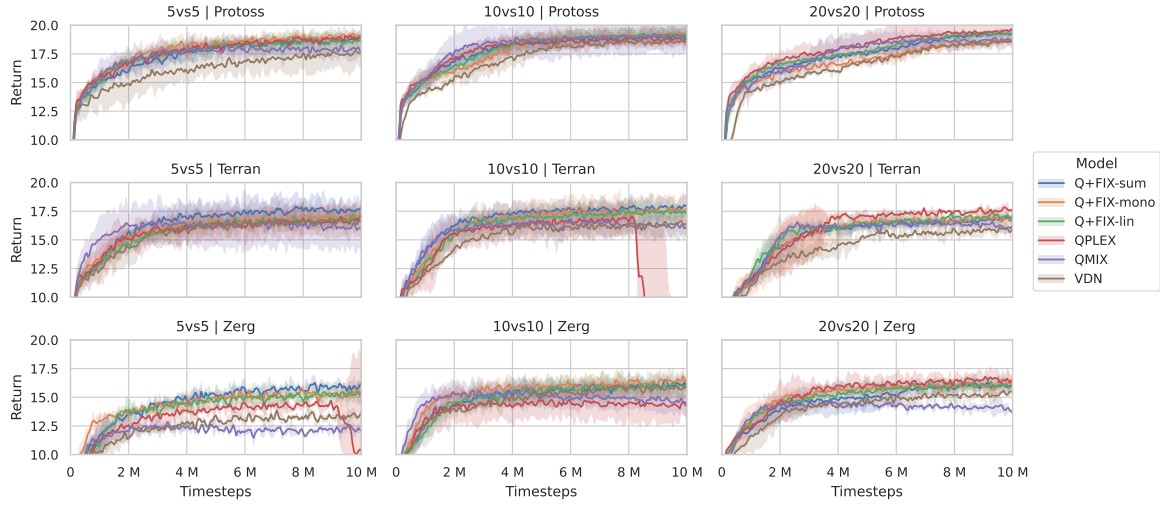

*Figure 2.* SMACv2 mean returns and bootstrapped confidence intervals.

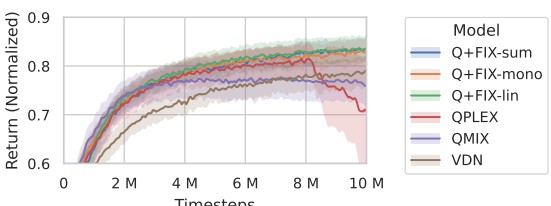

*Figure 3.* SMACv2 mean (normalized) aggregate returns and bootstrapped confidence intervals. Aggregates are normalized via $\tilde{G}_i \doteq \frac{G_i - \min_k G_k}{\max_k G_k - \min_k G_k} \in [0, 1]$, where $\{G_i\}_i$ is the total set of returns logged by all models in all scenarios.

as a consequence of the simpler minimalist structure.

Q+FIX-sum, Q+FIX-mono, and Q+FIX-lin achieve similar learning performances in most cases, with only minor differences across scenarios. Overall, Q+FIX-sum may be slightly outperforming other variants in some scenarios (`Terran-5vs5`, `Zerg-5vs5`), possibly an indication that a simpler compositions are preferable, so long as the full IGM-complete space is accessible.

The normalized aggregate returns in Figure 3 provide more accurate estimations of expected performance due to the larger sample size (27 total runs per model), and show more clearly the trends discussed above. With these aggregate results, it becomes more clear that, even ignoring the unstable convergence of QPLEX, the Q+FIX variants all manage to at least mildly outperform QPLEX. These results demonstrate that Q+FIX succeeds in enhancing the native performances of VDN and QMIX fixees, and lifts them to a similar level as QPLEX while maintaining more stable convergence. Finally, Table 1 shows that Q+FIX (especially Q+FIX-sum and Q+FIX-lin) is able to achieve these performances while

using the smallest mixing network by a significant margin.

## 6. Conclusions

In recent years, value function decomposition methods that employ the CTDE training paradigm for MARL have risen to state-of-the-art status, achieving significant learning performance benefits in cooperative multi-agent control problems. Such methods are often centered around the IGM property, and recent work has focused on developing models that are able to represent the entire IGM-complete function class. When put under scrutiny, most such methods have failed at that objective, and QPLEX represents the singular exception. However, QPLEX only represents a single instance in the space of models that achieve the IGM-complete function class, and whether other better options exist remained was an open question to explore.

In this work, we have advanced our understanding of the IGM-complete function class by proposing a minimal formulation of the IGM property that is directly implementable. Inspired by such formulation, we were able to naturally derive QFIX, a novel family of value function decomposition methods that enhance the representation capabilities of prior models via a simple manipulation of their outputs. As a result, we are able to implement a number of IGM-complete models that are significantly simpler than QPLEX. Our empirical evaluation on SMACv2 demonstrates that our QFIX methods succeed in both enhancing the performance of prior methods like VDN and QMIX, and achieving better convergence properties than QPLEX while needing a fraction of the parameters. Our contribution not only represents a novel approach that performs well, but also opens the door for new methods based on the QFIX framework.

## Impact Statement

This paper presents work whose goal is to advance the field of Multi-Agent Reinforcement Learning. There are many potential societal consequences of our work, none which we feel must be specifically highlighted here.

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

# A. Proofs

## A.1. Proof of Lemma 4.2

*Proof.* For any given joint history $h$, let $a_i^* = \text{argmax}_{a_i} Q_i(h_i, a_i)$ denote the maximal action according to the individual utilities, and $a^* = (a_1^*, \ldots, a_N^*)$ the joint action constructed by those individual actions.

For this joint action $a^*$, the corresponding advantage utilities are zero $\forall i \, (u_i^* = 0)$ by definition, and

$$Q_{\text{IGM}}(h, a^*) = w(h, a^*) \underbrace{f(u_1^*, \ldots, u_N^*)}_{=0} + b(h)$$

$$= b(h) \,. \tag{34}$$

For any other non-maximal action $a$, we have at least one strictly negative utility $\exists i \, (u_i < 0)$, and

$$Q_{\text{IGM}}(h, a) = \underbrace{w(h, a)}_{>0} \underbrace{f(u_1, \ldots, u_N)}_{<0} + b(h)$$

$$< b(h) \,. \tag{35}$$

Therefore $a^* = \text{argmax}_a Q_{\text{IGM}}(h, a)$, and the actions that maximize the individual utilities also maximize the joint value.

$\square$

## A.2. Proof of Theorem 4.3

*Proof.* Let us denote the function class of $Q_{\text{IGM}}$ as $\mathcal{FC}(Q_{\text{IGM}})$, and the IGM-complete function class as $\mathcal{FC}_{\text{IGM}}$. We prove the equivalence $\mathcal{FC}(Q_{\text{IGM}}) = \mathcal{FC}_{\text{IGM}}$ in two steps:

1. $Q \in \mathcal{FC}(Q_{\text{IGM}}) \implies Q \in \mathcal{FC}_{\text{IGM}}$, i.e., $Q_{\text{IGM}}$ satisfies IGM,

2. $Q \in \mathcal{FC}_{\text{IGM}} \implies Q \in \mathcal{FC}(Q_{\text{IGM}})$, i.e., any function that satisfies IGM can be represented by $Q_{\text{IGM}}$.

**Step 1.** $Q \in \mathcal{FC}(Q_{\text{IGM}}) \implies Q \in \mathcal{FC}_{\text{IGM}}$ follows directly from Lemma 4.2.

**Step 2.** Let $Q_i(h_i, a_i)$ and $Q(h, a)$ denote an arbitrary set of individual and joint values that satisfy IGM, i.e., $Q \in \mathcal{FC}_{\text{IGM}}$. Let us denote the usual corresponding values and advantages as follows,

$$V_i(h_i) = \max_{a_i} Q_i(h_i, a_i) \,, \qquad A_i(h_i, a_i) = Q_i(h_i, a_i) - V_i(h_i) \,, \tag{36}$$

$$V(h) = \max_a Q(h, a) \,, \qquad A(h, a) = Q(h, a) - V(h) \,, \tag{37}$$

with the usual shorthand $q_i = Q_i(h_i, a_i)$ and $v_i = V_i(h_i)$, and $u_i = A_i(h_i, a_i)$.

For any $f$ that satisfies the requirements of Equation (19), let $w$ and $b$ be defined as follows,

$$b(h) = V(h) \,, \tag{38}$$

$$w(h, a) = \begin{cases} \frac{A(h, a)}{f(u_1, \ldots, u_N)} \,, & \text{if } f(u_1, \ldots, u_N) \neq 0 \,, \\ \text{any value} \,, & \text{otherwise} \,. \end{cases} \tag{39}$$

For any given joint history $h$, let $a_i^* = \text{argmax}_{a_i} Q_i(h_i, a_i)$ denote the maximal action according to the individual utilities, and $a^* = (a_1^*, \ldots, a_N^*)$ the corresponding joint action. Given that $Q$ satisfies IGM by assumption, we have $a^* = \text{argmax}_a Q(h, a)$, and $Q(h, a^*) = \max_a Q(h, a) = V(h)$.

For this joint action $\boldsymbol{a}^*$, the corresponding individual advantage utilities are zero $\forall i \, (u_i = 0)$ by definition, and

$$
\begin{aligned}
Q_{\text{IGM}}(\boldsymbol{h}, \boldsymbol{a}^*) &= w(\boldsymbol{h}, \boldsymbol{a}^*)f(u_1, \ldots, u_N) + b(\boldsymbol{h}) \\
&= w(\boldsymbol{h}, \boldsymbol{a}^*) \underbrace{f(0, \ldots, 0)}_{=0} + b(\boldsymbol{h}) \\
&= V(\boldsymbol{h}) \\
&= Q(\boldsymbol{h}, \boldsymbol{a}^*).
\end{aligned}
\tag{40}
$$

For any other non-maximal action $\boldsymbol{a}^\dagger$, we have at least one strictly negative utility $\exists i \, (u_i < 0)$, and

$$
\begin{aligned}
Q_{\text{IGM}}(\boldsymbol{h}, \boldsymbol{a}^\dagger) &= w(\boldsymbol{h}, \boldsymbol{a}^\dagger)f(u_1, \ldots, u_N) + b(\boldsymbol{h}) \\
&= \frac{A(\boldsymbol{h}, \boldsymbol{a}^\dagger)}{f(u_1, \ldots, u_N)} f(u_1, \ldots, u_N) + V(\boldsymbol{h}) \\
&= A(\boldsymbol{h}, \boldsymbol{a}^\dagger) + V(\boldsymbol{h}) \\
&= Q(\boldsymbol{h}, \boldsymbol{a}^\dagger).
\end{aligned}
\tag{41}
$$

In either case, $Q_{\text{IGM}}(\boldsymbol{h}, \boldsymbol{a}) = Q(\boldsymbol{h}, \boldsymbol{a})$ for all joint histories and actions. Therefore $Q \in \mathcal{FC}_{\text{IGM}} \implies Q \in \mathcal{FC}(Q_{\text{IGM}})$.
$\qquad \square$

### A.3. Proof of Theorem 4.4

*Proof.* Equation (22) satisfies the form and requirements of Equation (19). Therefore, IGM follows from Lemma 4.2. Given $w$ and $b$ models that are sufficiently expressive, IGM-completeness follows from Theorem 4.3 $\qquad \square$

### A.4. Proof of Theorem 4.5

*Proof.* QFIX-lin is a monotonic function of individual advantages and therefore satisfies IGM. QFIX-lin is also a generalization of QFIX-sum, therefore its function class is a superset of the QFIX-sum function class, which is the IGM-complete function class. Therefore, QFIX-lin can represent all models that satisfy IGM, and none of those that do not satisfy IGM. $\quad \square$

### A.5. Proof of Theorem 4.6

*Proof.* $(w(\boldsymbol{h}, \boldsymbol{a}) + 1) > 0$ satisfies the positivity constraint of QFIX. Therefore, Theorem 4.4 applies. $\qquad \square$

## B. Derivations

This section contains explicit long-form derivations that had to be removed from the main document due to space limitations. Appendices B.1 and B.2 contain the maximal value $V(\boldsymbol{h})$ and advantage $A(\boldsymbol{h}, \boldsymbol{a})$ for VDN and QMIX. Appendices B.3 and B.4 contain the derivation for QFIX-sum and QFIX-mono. Appendices B.5 to B.7 contain the derivation for Q+FIX-sum, Q+FIX-mono, and Q+FIX-lin.

**B.1. VDN Maximal Values $\hat{V}_{\text{MIX}}(\boldsymbol{h})$ and Advantages $\hat{A}_{\text{MIX}}(\boldsymbol{h}, \boldsymbol{a})$**

As a reminder, VDN action-values are defined as $\hat{Q}_{\text{VDN}}(\boldsymbol{h}, \boldsymbol{a}) \doteq \sum_i \hat{Q}_i(h_i, a_i)$. Due to the the linear (monotonic) mixing structure, the joint maximal values $\hat{V}_{\text{VDN}}(\boldsymbol{h})$ can be expressed as the sum of the individual maximal values,

$$
\begin{aligned}
\hat{V}_{\text{VDN}}(\boldsymbol{h}) &\doteq \max_{\boldsymbol{a}} \hat{Q}_{\text{VDN}}(\boldsymbol{h}, \boldsymbol{a}) \\
&= \max_{\boldsymbol{a}} \sum_i \hat{Q}_i(h_i, a_i) \\
&= \max_{a_1, \ldots, a_N} \sum_i \hat{Q}_i(h_i, a_i) \\
&= \sum_i \max_{a_i} \hat{Q}_i(h_i, a_i) && \text{(monotonicity)} \\
&= \sum_i \hat{V}_i(h_i),
\end{aligned}
\tag{42}
$$

and the joint advantages $\hat{A}_{\text{VDN}}(\boldsymbol{h}, \boldsymbol{a})$ can be expressed as the sum of the individual advantages,

$$
\begin{aligned}
\hat{A}_{\text{VDN}}(\boldsymbol{h}, \boldsymbol{a}) &\doteq \hat{Q}_{\text{VDN}}(\boldsymbol{h}, \boldsymbol{a}) - \hat{V}_{\text{VDN}}(\boldsymbol{h}) \\
&= \sum_i \hat{Q}_i(h_i, a_i) - \sum_i \hat{V}_i(h_i) \\
&= \sum_i \hat{Q}_i(h_i, a_i) - \hat{V}_i(h_i) \\
&= \sum_i \hat{A}_i(h_i, a_i).
\end{aligned}
\tag{43}
$$

**B.2. QMIX Maximal Values $\hat{V}_{\text{MIX}}(\boldsymbol{h})$ and Advantages $\hat{A}_{\text{MIX}}(\boldsymbol{h}, \boldsymbol{a})$**

As a reminder, QMIX action-values are defined as $\hat{Q}_{\text{MIX}}(\boldsymbol{h}, \boldsymbol{a}) \doteq f_{\text{mono}}(q_1, \ldots, q_N)$. Due to the monotonic mixing structure, the joint maximal values $\hat{V}_{\text{MIX}}(\boldsymbol{h})$ can be expressed as the monotonic mixing of the individual maximal values,

$$
\begin{aligned}
\hat{V}_{\text{MIX}}(\boldsymbol{h}) &\doteq \max_{\boldsymbol{a}} \hat{Q}_{\text{MIX}}(\boldsymbol{h}, \boldsymbol{a}) \\
&= \max_{\boldsymbol{a}} f_{\text{mono}}(q_1, \ldots, q_N) \\
&= \max_{a_1, \ldots, a_N} f_{\text{mono}}\left(\hat{Q}_1(h_1, a_1), \ldots, \hat{Q}_N(h_N, a_N)\right) \\
&= f_{\text{mono}}\left(\max_{a_1} \hat{Q}_1(h_1, a_1), \ldots, \max_{a_N} \hat{Q}_N(h_N, a_N)\right) && \text{(monotonicity)} \\
&= f_{\text{mono}}\left(\hat{V}_1(h_1), \ldots, \hat{V}_N(h_N)\right) \\
&= f_{\text{mono}}(v_1, \ldots, v_N),
\end{aligned}
\tag{44}
$$

and the joint advantages $\hat{A}_{\text{MIX}}(\boldsymbol{h}, \boldsymbol{a})$ can be expressed as the corresponding difference,

$$
\begin{aligned}
\hat{A}_{\text{MIX}}(\boldsymbol{h}, \boldsymbol{a}) &\doteq \hat{Q}_{\text{MIX}}(\boldsymbol{h}, \boldsymbol{a}) - \hat{V}_{\text{MIX}}(\boldsymbol{h}) \\
&= f_{\text{mono}}(q_1, \ldots, q_N) - f_{\text{mono}}(v_1, \ldots, v_N).
\end{aligned}
\tag{45}
$$

**B.3. QFIX-sum**

QFIX-sum is an instance of QFIX based on VDN as fixee model, $\hat{Q}_{\text{fixee}}(\boldsymbol{h}, \boldsymbol{a}) = \hat{Q}_{\text{VDN}}(\boldsymbol{h}, \boldsymbol{a})$. From Equation (43), we have that the VDN joint advantage is given as the sum of individual advantages (hence the "-sum" suffix). Therefore, QFIX-sum is simply obtained as

$$
\begin{aligned}
\hat{Q}_{\text{FIX-sum}}(\boldsymbol{h}, \boldsymbol{a}) &\doteq w(\boldsymbol{h}, \boldsymbol{a}) \hat{A}_{\text{VDN}}(\boldsymbol{h}, \boldsymbol{a}) + b(\boldsymbol{h}) \\
&= w(\boldsymbol{h}, \boldsymbol{a}) \sum_i \hat{A}_i(h_i, a_i) + b(\boldsymbol{h}).
\end{aligned}
\tag{46}
$$

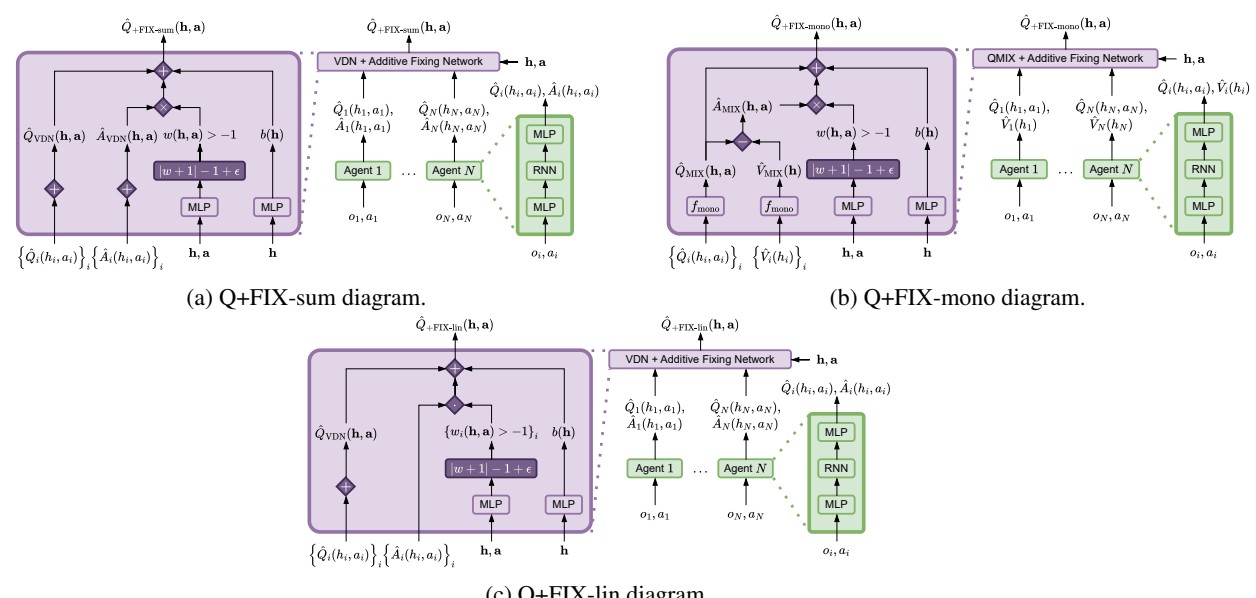

*Figure 4.* Specialized diagrams for Q+FIX-sum, Q+FIX-mono, and Q+FIX-lin.

### B.4. QFIX-mono

QFIX-mono is an instance of QFIX based on QMIX as fixee model, $\hat{Q}_{\text{fixee}}(\boldsymbol{h}, \boldsymbol{a}) = \hat{Q}_{\text{MIX}}(\boldsymbol{h}, \boldsymbol{a})$. From Equation (45), we have that the QMIX advantage is given as a difference between monotonic compositions of individual utilities (hence the "-mono" suffix). Therefore, QFIX-mono is simply obtained as

$$\hat{Q}_{\text{FIX-mono}}(\boldsymbol{h}, \boldsymbol{a}) \doteq w(\boldsymbol{h}, \boldsymbol{a})\hat{A}_{\text{MIX}}(\boldsymbol{h}, \boldsymbol{a}) + b(\boldsymbol{h})$$
$$= w(\boldsymbol{h}, \boldsymbol{a})(f_{\text{mono}}(q_1, \ldots, q_N) - f_{\text{mono}}(v_1, \ldots, v_N)) + b(\boldsymbol{h}) \,. \tag{47}$$

### B.5. Q+FIX-sum

Q+FIX-sum is an instance of Q+FIX based on VDN as fixee model, $\hat{Q}_{\text{fixee}}(\boldsymbol{h}, \boldsymbol{a}) = \hat{Q}_{\text{VDN}}(\boldsymbol{h}, \boldsymbol{a})$ and $\hat{A}_{\text{fixee}}(\boldsymbol{h}, \boldsymbol{a}) = \hat{A}_{\text{VDN}}(\boldsymbol{h}, \boldsymbol{a})$, also equivalent to the additive formulation of QFIX-sum. Therefore, Q+FIX-sum is simply obtained as

$$\hat{Q}_{+\text{FIX-sum}} \doteq \hat{Q}_{\text{VDN}}(\boldsymbol{h}, \boldsymbol{a}) + w(\boldsymbol{h}, \boldsymbol{a})\hat{A}_{\text{VDN}}(\boldsymbol{h}, \boldsymbol{a}) + b(\boldsymbol{h})$$
$$\doteq \sum_i \hat{Q}_i(\boldsymbol{h}, \boldsymbol{a}) + w(\boldsymbol{h}, \boldsymbol{a}) \sum_i \hat{A}_i(\boldsymbol{h}, \boldsymbol{a}) + b(\boldsymbol{h}) \,. \tag{48}$$

Figure 4a shows a graphical diagram for Q+FIX-sum.

### B.6. Q+FIX-mono

Q+FIX-mono is an instance of Q+FIX based on QMIX as fixee model, $\hat{Q}_{\text{fixee}}(\boldsymbol{h}, \boldsymbol{a}) = \hat{Q}_{\text{MIX}}(\boldsymbol{h}, \boldsymbol{a})$ and $\hat{A}_{\text{fixee}}(\boldsymbol{h}, \boldsymbol{a}) = \hat{A}_{\text{MIX}}(\boldsymbol{h}, \boldsymbol{a})$, also equivalent to the additive formulation of QFIX-mono. Therefore, Q+FIX-mono is simply obtained as

$$\hat{Q}_{+\text{FIX-mono}} \doteq \hat{Q}_{\text{VDN}}(\boldsymbol{h}, \boldsymbol{a}) + w(\boldsymbol{h}, \boldsymbol{a})\hat{A}_{\text{VDN}}(\boldsymbol{h}, \boldsymbol{a}) + b(\boldsymbol{h})$$
$$\doteq f_{\text{mono}}(q_1, \ldots, q_N) + w(\boldsymbol{h}, \boldsymbol{a})(f_{\text{mono}}(q_1, \ldots, q_N) - f_{\text{mono}}(v_1, \ldots, v_N)) + b(\boldsymbol{h}) \,. \tag{49}$$

Figure 4b shows a graphical diagram for Q+FIX-mono.

### B.7. Q+FIX-lin

Q+FIX-lin is the additive formulation of QFIX-lin. Just as QFIX-lin is not formally a member of the QFIX family, but rather a generalization of QFIX-sum, so is Q+FIX-lin not formally a member of Q+FIX, but rather a generalization of Q+FIX-sum.

Given that QFIX-lin is obtained by introducing per-agent weights $w_i(\boldsymbol{h}, \boldsymbol{a})$, Q+FIX-lin is simply obtained as

$$\hat{Q}_{+\text{FIX-lin}} \doteq \sum_i \hat{Q}_i(h_i, a_i) + \sum_i w_i(\boldsymbol{h}, \boldsymbol{a})\hat{A}_i(h_i, a_i) + b(\boldsymbol{h}).$$

Figure 4c shows a graphical diagram for Q+FIX-lin.

## C. Why Detaching the Advantages Helps Q+FIX

First, we note that the gradients $\nabla_{\theta_i}\hat{Q}_{+\text{FIX}}(\boldsymbol{h}, \boldsymbol{a})$ when the advantages *are not* detached are as follows,

$$
\begin{aligned}
&\nabla_{\theta_i}\hat{Q}_{+\text{FIX}}(\boldsymbol{h}, \boldsymbol{a}) \\
&= \nabla_{\theta_i}\hat{Q}_{\text{fixee}}(\boldsymbol{h}, \boldsymbol{a}) + w(\boldsymbol{h}, \boldsymbol{a})\nabla_{\theta_i}\hat{A}_{\text{fixee}}(\boldsymbol{h}, \boldsymbol{a}) \\
&= \nabla_{\theta_i}\hat{V}_{\text{fixee}}(\boldsymbol{h}) + (w(\boldsymbol{h}, \boldsymbol{a}) + 1)\nabla_{\theta_i}\hat{A}_{\text{fixee}}(\boldsymbol{h}, \boldsymbol{a}).
\end{aligned}
\tag{50}
$$

It seems plausible that there may be poor values of $w(\boldsymbol{h}, \boldsymbol{a})$ that could result in degenerate gradient signals. For example, a low fixing weight $w(\boldsymbol{h}, \boldsymbol{a}) \approx -1$ results in a dampened gradient $\nabla_{\theta_i}\hat{Q}_{+\text{FIX}}(\boldsymbol{h}, \boldsymbol{a}) \approx \nabla_{\theta_i}\hat{V}_{\text{fixee}}(\boldsymbol{h})$, that is notably independent on actions. On the other end of the spectrum, a very large fixing weight $w(\boldsymbol{h}, \boldsymbol{a}) \gg -1$ results in a gradient that is dominated by the highly-weighted advantage component, overcoming the value component, $\nabla_{\theta_i}\hat{Q}_{+\text{FIX}}(\boldsymbol{h}, \boldsymbol{a}) \approx w(\boldsymbol{h}, \boldsymbol{a})\nabla_{\theta_i}\hat{A}_{\text{fixee}}(\boldsymbol{h}, \boldsymbol{a})$. On each end of the spectrum, the gradient will propagate almost exclusively through the values $\nabla_{\theta_i}\hat{V}_{\text{fixee}}(\boldsymbol{h})$ or through the advantages $\nabla_{\theta_i}\hat{A}_{\text{fixee}}(\boldsymbol{h}, \boldsymbol{a})$.

On the other hand, the gradients $\nabla_{\theta_i}\hat{Q}_{+\text{FIX}}(\boldsymbol{h}, \boldsymbol{a})$ when the advantages *are* detached are as follows,

$$
\begin{aligned}
\nabla_{\theta_i}\hat{Q}_{+\text{FIX}}(\boldsymbol{h}, \boldsymbol{a}) &= \nabla_{\theta_i}\hat{Q}_{\text{fixee}}(\boldsymbol{h}, \boldsymbol{a}) \\
&= \nabla_{\theta_i}\hat{V}_{\text{fixee}}(\boldsymbol{h}) + \nabla_{\theta_i}\hat{A}_{\text{fixee}}(\boldsymbol{h}, \boldsymbol{a}),
\end{aligned}
\tag{51}
$$

and are unaffected by the fixing structure, equally dependent on the value and advantage components of $\hat{Q}_{\text{fixee}}(\boldsymbol{h}, \boldsymbol{a})$.

## D. Stateful QFIX

For simplicity, we assume a stateless fixee $\hat{Q}_{\text{fixee}}(\boldsymbol{h}, \boldsymbol{a})$, although these results can be easily extended to stateful fixees $\hat{Q}_{\text{fixee}}(\boldsymbol{h}, s, \boldsymbol{a})$ under mild conditions.

### D.1. History-State QFIX

**IGM**  In the case of history-state QFIX, as defined by

$$\hat{Q}_{\text{FIX}}(\boldsymbol{h}, s, \boldsymbol{a}) \doteq w(\boldsymbol{h}, s, \boldsymbol{a})\hat{A}_{\text{fixee}}(\boldsymbol{h}, \boldsymbol{a}) + b(\boldsymbol{h}, s),\tag{52}$$

where $w(\boldsymbol{h}, s, \boldsymbol{a}) > 0$, we first show that $\hat{Q}_{\text{FIX}}(\boldsymbol{h}, s, \boldsymbol{a})$ satisfies stateful-IGM. We employ the same methodology used by Marchesini et al. (2024), whereby we show that the presence of the state is able to alter the values of $\hat{Q}_{\text{FIX}}(\boldsymbol{h}, s, \boldsymbol{a})$, but not the identity of the corresponding maximal action. For that purpose, let $\boldsymbol{a}^* = \operatorname{argmax}\hat{A}_{\text{fixee}}(\boldsymbol{h}, \boldsymbol{a})$ be the maximal action of the fixee. For that action $\boldsymbol{a}^*$, we have that

$$\hat{Q}_{\text{FIX}}(\boldsymbol{h}, s, \boldsymbol{a}^*) = w(\boldsymbol{h}, s, \boldsymbol{a}^*)\underbrace{\hat{A}_{\text{fixee}}(\boldsymbol{h}, \boldsymbol{a}^*)}_{=0} + b(\boldsymbol{h}, s)\tag{53}$$

$$= b(\boldsymbol{h}, s),\tag{54}$$

whereas for any other non-maximal action $\boldsymbol{a}$, we have

$$\hat{Q}_{\text{FIX}}(\boldsymbol{h}, s, \boldsymbol{a}) = \underbrace{w(\boldsymbol{h}, s, \boldsymbol{a})}_{>0}\underbrace{\hat{A}_{\text{fixee}}(\boldsymbol{h}, \boldsymbol{a}^*)}_{<0} + b(\boldsymbol{h}, s)\tag{55}$$

$$< b(\boldsymbol{h}, s).\tag{56}$$

Therefore, the action $\boldsymbol{a}^*$ that maximizes the fixee $\hat{Q}_{\text{fixee}}(\boldsymbol{h}, \boldsymbol{a})$ also maximizes the stateful QFIX $\hat{Q}_{\text{FIX}}(\boldsymbol{h}, s, \boldsymbol{a})$ regardless of the state. Since the fixee is assumed to satisfy IGM, then the same set of individual actions maximize the individual utilities $\hat{Q}_i(h_i, a_i)$, therefore QFIX satisfies stateful-IGM.

**IGM-Completeness** This proof takes on a similar form to that for Theorem 4.3, although we proceed less formally. We need to prove that any stateful value function $Q(\boldsymbol{h}, s, \boldsymbol{a})$ that satisfies stateful-IGM can be represented via history-state QFIX. Let $V(\boldsymbol{h}, s) \doteq \max_{\boldsymbol{a}} \mathbb{E}_{s|\boldsymbol{h}}[Q(\boldsymbol{h}, s, \boldsymbol{a})]$ and $A(\boldsymbol{h}, s, \boldsymbol{a}) \doteq Q(\boldsymbol{h}, s, \boldsymbol{a}) - V(\boldsymbol{h}, s)$. Note the distinction between $Q(\boldsymbol{h}, s, \boldsymbol{a})$, the stateful-IGM-compliant value we aim to model, and $\hat{Q}_{\text{fixee}}(\boldsymbol{h}, \boldsymbol{a})$, the fixee we attempt to fix. To that end, let $w$ and $b$ be defined as follows,

$$b(\boldsymbol{h}, s) = V(\boldsymbol{h}, s), \tag{57}$$

$$w(\boldsymbol{h}, s, \boldsymbol{a}) = \begin{cases} \frac{A(\boldsymbol{h}, s, \boldsymbol{a})}{\hat{A}_{\text{fixee}}(\boldsymbol{h}, \boldsymbol{a})}, & \text{if } \hat{A}_{\text{fixee}}(\boldsymbol{h}, \boldsymbol{a}) \neq 0, \\ \text{any value}, & \text{otherwise}. \end{cases} \tag{58}$$

For any given joint history $\boldsymbol{h}$, let $a_i^* = \operatorname{argmax}_{a_i} Q_i(h_i, a_i)$ denote the maximal action according to the individual utilities, and $\boldsymbol{a}^* = (a_1^*, \ldots, a_N^*)$ the corresponding joint action. Given that $Q(\boldsymbol{h}, s, \boldsymbol{a})$ satisfies stateful-IGM by assumption, we have $\boldsymbol{a}^* = \operatorname{argmax}_{\boldsymbol{a}} \mathbb{E}_{s|\boldsymbol{h}}[Q(\boldsymbol{h}, s, \boldsymbol{a})]$ and $Q(\boldsymbol{h}, s, \boldsymbol{a}^*) = V(\boldsymbol{h}, s)$.

For this joint action $\boldsymbol{a}^*$, the corresponding fixee advantage is zero by definition, and

$$\hat{Q}_{\text{FIX}}(\boldsymbol{h}, s, \boldsymbol{a}^*) = w(\boldsymbol{h}, s, \boldsymbol{a}^*) \underbrace{\hat{A}_{\text{fixee}}(\boldsymbol{h}, \boldsymbol{a}^*)}_{=0} + b(\boldsymbol{h}, s) \tag{59}$$

$$= b(\boldsymbol{h}, s) \tag{60}$$

$$= V(\boldsymbol{h}, s) \tag{61}$$

$$= Q(\boldsymbol{h}, s, \boldsymbol{a}^*). \tag{62}$$

For any other non-maximal action $\boldsymbol{a}^\dagger$, we have $\hat{A}_{\text{fixee}}(\boldsymbol{h}, \boldsymbol{a}^\dagger) < 0$, and

$$\hat{Q}_{\text{FIX}}(\boldsymbol{h}, s, \boldsymbol{a}^\dagger) = w(\boldsymbol{h}, s, \boldsymbol{a}^\dagger) \hat{A}_{\text{fixee}}(\boldsymbol{h}, \boldsymbol{a}^\dagger) + b(\boldsymbol{h}, s) \tag{63}$$

$$= \frac{A(\boldsymbol{h}, s, \boldsymbol{a}^\dagger)}{\hat{A}_{\text{fixee}}(\boldsymbol{h}, \boldsymbol{a}^\dagger)} \hat{A}_{\text{fixee}}(\boldsymbol{h}, \boldsymbol{a}^\dagger) + V(\boldsymbol{h}, s) \tag{64}$$

$$= A(\boldsymbol{h}, s, \boldsymbol{a}^\dagger) + V(\boldsymbol{h}, s) \tag{65}$$

$$= Q(\boldsymbol{h}, s, \boldsymbol{a}^\dagger). \tag{66}$$

In either case, $\hat{Q}_{\text{FIX}}(\boldsymbol{h}, s, \boldsymbol{a}) = Q(\boldsymbol{h}, s, \boldsymbol{a})$ for all joint histories, states, and joint actions.

### D.2. State-Only QFIX

**IGM** In the case of state-only QFIX, as defined by

$$\hat{Q}_{\text{FIX}}(\boldsymbol{h}, s, \boldsymbol{a}) \doteq w(s, \boldsymbol{a}) \hat{A}_{\text{fixee}}(\boldsymbol{h}, \boldsymbol{a}) + b(s), \tag{67}$$

where $w(s, \boldsymbol{a}) > 0$, we first show that $\hat{Q}_{\text{FIX}}(\boldsymbol{h}, s, \boldsymbol{a})$ satisfies stateful-IGM. We employ the same methodology used above, whereby we show that the presence of the state is able to alter the values of $\hat{Q}_{\text{FIX}}(\boldsymbol{h}, s, \boldsymbol{a})$, but not the identity of the corresponding maximal action. For that purpose, let $\boldsymbol{a}^* = \operatorname{argmax} \hat{A}_{\text{fixee}}(\boldsymbol{h}, \boldsymbol{a})$ be the maximal action of the fixee. For that action $\boldsymbol{a}^*$, we have that

$$\hat{Q}_{\text{FIX}}(\boldsymbol{h}, s, \boldsymbol{a}^*) = w(s, \boldsymbol{a}^*) \underbrace{\hat{A}_{\text{fixee}}(\boldsymbol{h}, \boldsymbol{a}^*)}_{=0} + b(s) \tag{68}$$

$$= b(s), \tag{69}$$

whereas for any other non-maximal action $\boldsymbol{a}$, we have

$$\hat{Q}_{\text{FIX}}(\boldsymbol{h}, s, \boldsymbol{a}) = \underbrace{w(s, \boldsymbol{a})}_{>0} \underbrace{\hat{A}_{\text{fixee}}(\boldsymbol{h}, \boldsymbol{a}^*)}_{<0} + b(s) \tag{70}$$

$$< b(s). \tag{71}$$

Therefore, the action $\boldsymbol{a}^*$ that maximizes the fixee $\hat{Q}_{\text{fixee}}(\boldsymbol{h}, \boldsymbol{a})$ also maximizes the stateful QFIX $\hat{Q}_{\text{FIX}}(\boldsymbol{h}, s, \boldsymbol{a})$ regardless of the state. Since the fixee is assumed to satisfy IGM, then the same set of individual actions maximize the individual utilities $\hat{Q}_i(h_i, a_i)$, therefore state-only QFIX satisfies stateful-IGM.

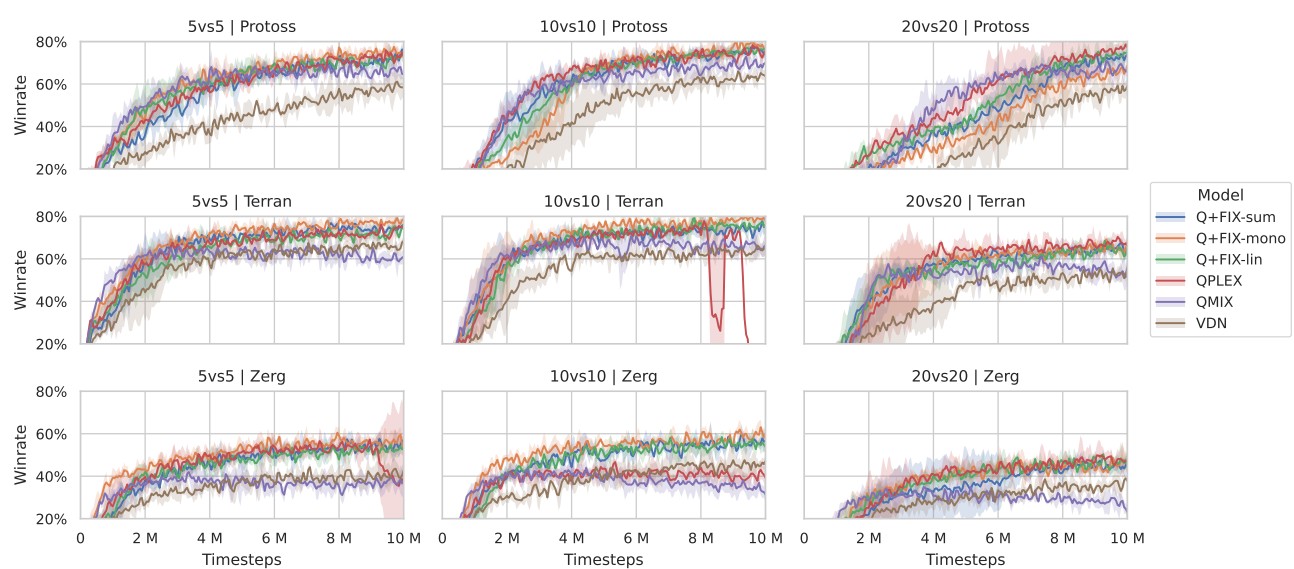

*Figure 5.* SMACv2 mean winrates and bootstrapped confidence intervals.

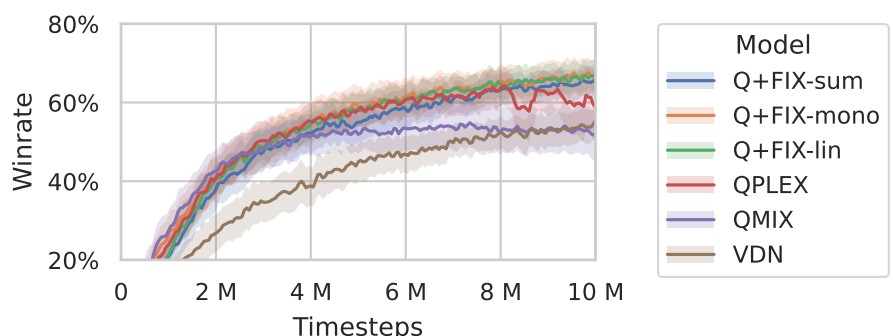

*Figure 6.* SMACv2 mean (normalized) aggregate winrates and bootstrapped confidence intervals.

**IGM-Completeness**  In contrast to history-state QFIX, we are not able to use the same proof to show that state-only QFIX satisfies IGM-completeness.

## E. Additional Winrate Results

In this section, we show additional results based on the winrate metric. As with the return-based results, we show the learning performance for each model and scenario in Figure 5, and the aggregate winrate across scenarios in Figure 6.

**Winrates vs Returns**  As mentioned in the main document, the winrate and return metrics induce correlated but notably different orderings over the evaluated methods. Comparing Figures 2 and 5, this is notable by the following non-exhaustive examples:

- In `Terran-5vs5`,
    - Return implies Q+FIX-sum $\succ$ Q+FIX-mono.
    - Winrate implies Q+FIX-sum $\prec$ Q+FIX-mono.

- In `Zerg-5vs5`,
    - Return implies Q+FIX-sum $\succ$ Q+FIX-mono $\approx$ Q+FIX-lin.

– Winrate implies Q+FIX-sum ≈ Q+FIX-mono ≈ Q+FIX-lin.

- In `Zerg-10vs10`,

    – Return implies VDN ≈ Q+FIX.
    – Winrate implies VDN ≺ Q+FIX.

- In `Protoss-20vs20`,

    – Return implies VDN ≈ Q+FIX-mono.
    – Winrate implies VDN ≺ Q+FIX-mono.

- In `Terran-10vs10`, the return of QPLEX drops significantly around the $9M$ timestep mark, whereas its winrate is able to recover temporarily, indicating that high winrates are achievable even with low returns.

Comparing the final performances in Figures 3 and 6,

- Return implies VDN ≺ QMIX ≺ QPLEX.

- Winrate implies QPLEX ≺ VDN ≈ QMIX.

**Winrate Results**   Despite this notable and concerning difference between returns and winrates as evaluation metrics, the winrate-based evaluation arrives to largely the same conclusions as the return-based one in the main document.

As in the return-based results, VDN fails to be a competitive baseline on its own for most scenarios, likely due to the well-known limited representation. Fixing VDN via Q+FIX-sum, we are able to overcome this limitation (as noted by the performance gap between VDN and Q+FIX-sum), expanding its representation space and reaching SOTA performance.

As in the return-based results, QMIX sometimes exhibits fast initial learning speeds, albeit often to a sub-competitive final performance (`Protoss-5vs5`, `Terran-5vs5`, `Terran-10vs10`, `Zerg-10vs10`, `Terran-20vs20`, `Zerg-20vs20`), again a likely consequence of its limited representation. Fixing QMIX via Q+FIX-mono, we are often able to exploit the initial learning speeds and complement them with improved performance at convergence reaching SOTA performance.

Compared to return-based results, QPLEX appears less competitive, and performs very well in fewer scenarios (`Protoss-20vs20`, `Terran-20vs20`, `Zerg-20vs20`), and underperforms in more (`Terran-5vs5`, `Zerg-10vs10`), and exhibits the same troubling convergence instabilities as well (`Zerg-5vs5`, `Terran-10vs10`). Q+FIX-lin, as the simplified variant inspired by QPLEX, manages to avoid such convergence instabilities, plausibly as a consequence of the simpler minimalist structure.

As in the return-based results, Q+FIX-sum, Q+FIX-mono, and Q+FIX-lin achieve similar learning performances in most cases, with only minor differences across scenarios. Compared to the return-based results, it is Q+FIX-mono that may be slightly outperforming other variants in some scenarios (`Terran-5vs5`, `Zerg-5vs5`).

The normalized aggregate returns in Figure 3 largely confirm the trends discussed above. Despite the concerning difference between the return and winrate metrics, both demonstrate that Q+FIX succeeds in enhancing the native performances of VDN and QMIX fixees, and lifts them to a similar level as QPLEX while maintaining more stable convergence.

