# OpenReview forum: "Fixing Value Function Decomposition for Multi-Agent Reinforcement Learning"
_ICML.cc/2025/Conference — Submitted to ICML 2025_

### Official Review · Reviewer_Cnks · 2025-02-25

**Overall Recommendation:** 2

**Summary:**

The paper studies the individual-global max (IGM) principle in model-based multi-agent reinforcement learning (MARL). They introduce a novel characterization of function classes of value function approximators, referred to as IGM-complete. They show the equivalence between this class and a parameterization of the agents' critic, and use this parameterization to "fix" existing value-based method, an approach they refer to as QFix. Experiments show the utility of QFix.

## Update after rebuttal
I acknowledge that the authors answered quite a few of my concerns, but not all of them. I find the theoretical contribution of this paper insufficient, even after the rebuttal. The authors have shown me their empirical results in more detail, and I appreciate them. However, at this stage, I do not see this paper as a high-quality whole yet, and thus I stand by my weak reject.

**Claims And Evidence:**

It seems clear that the technical work done by the authors delivered stronger results in the targetted benchmarks. However, there are issues with the evidence provided by the authors for the benefits of the method, both theoretically and empirically.

Lines 157-160 (right). *“... also demonstrate that QPLEX satisfies Definition 3.2 and its function class is IGM-complete, given sufficiently expressive models wi(h), bi(h), and λi(h, a).”* I do not see such a statement in the QPLEX paper. They just show that IGM can be defined in two ways.

Line 162 (right). *“Practical implementations of value function decomposition methods often employ stateful joint values”* - such a claim should be heavily supported with citations; especially that using both state and history as arguments makes little sense.

Definition 4.1. There is a problem with it, and you are not the only ones to blame. It seems that this definition (and Def 3.3) don’t handle the case when $A(h,a) < 0$ but $A_i(h_i, a_i) = 0$ properly. Indeed, IGM is of less use if this case is possible. Thus, NONE of them are actually equivalent to IGM (Def 3.1) as it demands that a global argmax can be formed from local argmaxes: thus, we would need $A_i(h_i, a_i)=0 \implies A(h, a)=0$, but your definition misses it.

Equation (19). This paremeterizatio is interesting, but it makes the results Lemma 4.2 and Theorem 4.3 trivial at the same time. Yes, you can take $b(h)=V(h)$. But then, you can obviously define $f(u_1,...,u_N)=-\mathbf{1}(\exists u_i \neq 0)$, where $\mathbf{1}(\cdot)$ is an indicator function, and set $w(h,a)=-A(h,a)$. In the following claims, you say that *$Q_{IGM}$ is a minimal function class*. It is not clear what it means. If you mean that it requires as few parametric models as possible, my example above shows you that it is not true.

Line 539 (Appendix). I don’t think you can take *“any $f(\cdot)$ that satisfies Eq (19)”* as it involves $w(h,a)$ that you haven’t defined yet. That can be probably fixed by defining $f(\cdot)$ and w jointly as a pair of functions that satisfy Eq 19, although this may be a poor characterization.

Theorem 4.4. What do you mean by sufficiently expressive?
Also, I don’t think that you can make such a statement without unrealistic assumptions about your function class (like *it can represent any function*) or set constraints on the functions you try to approximate, like continuity. To see why, look at this example: if history is continuous, and $f(u_1, …, u_N) = |{u_i | u_i \neq 0}| =: N_{nonzero}$, then $w(h,a)=[Q(h,a) - b(h)] / N_{nonzero}$ is, in principle, discontinuous, and thus for sure you cannot implement it with a neural network.

**Essential References Not Discussed:**

N/A

**Experimental Designs Or Analyses:**

I think that the experiments are, in a great deal, sound and of good quality. However, I believe the authors should ablate QFix against the base methods (like QMix and QPLEX) with bigger networks, to isolate the effect of the fixing network from additional parameter count.

**Methods And Evaluation Criteria:**

The authors demonstrate the improvement of performance of the fixees with QFix across a range of tasks. However, as the fixing network is applied on top of (not instead) of base fixee networks, it is unclear if the benefit comes from the fixing mechanisms or from additional neural computation. I would need to see additional ablation experiments to be convinced of the utility of QFix.

**Other Comments Or Suggestions:**

Please attach the important assumptions to your theoretical work, so that your claims can be stated rigorously. Also, please ablate QFix against the fixees with larger networks, to isolate the effect of the fixing network. With these changes, I will consider raising my score.

**Other Strengths And Weaknesses:**

The paper is extensive, dense with information, and easy to follow. The experimental range is wide, and the figures are appealing.

**Questions For Authors:**

Line 67 (left). *“However, WQMIX appears to conflate the possibility of exploiting state information during centralized training (which is correct) with the goal of learning the decision process for a team of fully observable agents (which is incorrect).”* - what do you mean by this?

Line 58 (right). You missed the distribution p in your POMDP-defining tuple.

Line 68 (right). You define joint history space before defining history (line 75).

**Relation To Broader Scientific Literature:**

The contribution targetted by the authors is valid and impactful. Value function decomposition and IGM are important areas of study for MARL.

**Theoretical Claims:**

I checked the correctness of the proofs. With sorrow I confess that, as I described in Claims and Evidence, I do not find these results very strong, nor do I think their proofs are particularly rigorous.

---

> ### Author Rebuttal · Authors · 2025-04-01
>
> We thank the reviewer for the thorough feedback; we will use it to improve the clarity of the submission.
>
> # Direct Questions
>
> ## Re: WQMIX
>
> See our response to Reviewer `5gNj`.
> We will clarify this comment in the paper.
>
> # Other Comments
>
> ## Re: QPLEX statement
>
> The corresponding statement for QPLEX is Prop. 2 of [1].
>
> ## Re: stateful joint values
>
> All common implementations employ stateful joint values; this includes: the QMIX code [3], the QPLEX code [4], and the popular `pymarl2` code [5].
> The QMIX and QPLEX papers also repeatedly refer to joint history-state values (although their notation is also often inconsistent).
> We will add these more explicit references to the paper.
>
> We also note “using models of both state and history makes little sense” is a common misconception.
> The literature of RL [6,7], MARL [8,9], and value-decomposition [2] has shown in both theory and practice that (mis)using models of state (without history) can be highly problematic, and that models of history and state can be more appropriate.
> Seminal work in MARL is slowly adapting to this misconception, e.g., see Errata in [10].
>
> ## Re: Definition 4.1
>
> This is indeed a discrepancy between def 3.1 and defs 3.3, 4.1.
> However, the root cause is that def 3.1 prescribes a unique maximum, while defs 3.3, 4.1 don’t.
> If defs 3.3, 4.1 were combined with unique maxima, then all definitions would be equivalent.
> That said, the generalization to multiple maxima is useful, and we agree that defs 3.3, 4.1 should be clarified as requested.
> We will clarify this point in the paper, but we also note that this has no effect on the rest of the work, as QPLEX, QIGM and QFIX already enforce a double implication.
>
> ## Re: Eq (19), Lemma 4.2, Thm 4.3
>
> That lemma 4.2 and thm 4.3 follow so naturally from eq (19) is a strength and sign of simplicity; not a weakness.
> Note: the given example is a special case of our proof for one specific choice of $f$, whereas our thms are valid for any choice of $f$.
> The more general proof is necessary to derive QFIX by replacing $f$ with **any** IGM fixee.
>
> ## Re: Minimalism of QIGM
>
> Our claims of minimalism are informed both quantitatively (by model sizes, Table 1), and qualitatively (eqs (22,25) are simpler than eq (16)).
> Note: we never claim that QIGM represents a minimal (small) function class; only that it is a minimal (simple) formulation of the IGM class.
>
> ## Re: proof of Thm 4.3
>
> We understand the confusion.
> The text states "For any $f$ that satisfies the requirements of eq (19)"
> We mean the requirements associated with eq (19), not eq (19) itself, i.e., $f$ non-positive and zero iff all inputs are zero.
> In eq (19), $f$ and $w$ are not codependent.
> In line 539, $w$ is constructed as a function of $f$ only to prove IGM-completeness; we are not claiming that this is a necessary structure of $w$.
>
> ## Re: Sufficiently expressive models
>
> This is an appeal to the Universal Approximation Thm, and consistent with other proofs in the literature [1].
> We will clarify this point.
> See our response to Reviewer `zepP` for a detailed reply.
>
> ## Re: Ablation
>
> The proposed ablation is only feasible in one case:
> - VDN does not use a parameterized mixing model; the requested ablation is impossible for Q+FIX-{sum,lin}.
> - QPLEX is already IGM-complete and is never used as a fixee; the requested ablation is again impossible.
> - QMIX is the one case where the proposed ablation is feasible; we are working to run this ablation by making Q+FIX-mono fixee smaller and/or QMIX larger. If we get preliminary results within the discussion period, we will post them.
>
> However, note that the performances of all Q+FIX variants are comparable, yet Q+FIX-{sum,lin} employ the **smallest** mixers by far.
> This hints at our mixing structure as a core contributor of performance regardless of fixee size; we expect the additional ablation to also confirm this.
> Also see our response to Reviewer `zepP` for a related topic.
>
> # Rebuttal Summary
>
> We believe we have addressed all concerns and, aside from minor clarifications, we reaffirm the correctness and rigorousness of our theory and results.
> We hope the reviewer will reconsider their evaluation positively and that they will let us know of any further concern.
>
> 1. Wang et al. "QPLEX: Duplex Dueling Multi-Agent Q-Learning" ICLR 2021
> 2. Marchesini et al. "On Stateful Value Factorization in Multi-Agent Reinforcement Learning" AAMAS 2025
> 3. github.com/oxwhirl/pymarl
> 4. github.com/wjh720/QPLEX
> 5. github.com/benellis3/pymarl2
> 6. Baisero et al. "Unbiased Asymmetric Reinforcement Learning under Partial Observability" AAMAS 2022
> 7. Baisero et al. "Asymmetric DQN for Partially Observable Reinforcement Learning" UAI 2022
> 8. Lyu et al. "A Deeper Understanding of State-Based Critics in Multi-Agent Reinforcement Learning" AAAI 2022
> 9. Lyu et al. "On centralized critics in multi-agent reinforcement learning" JAIR 2023
> 10. Foerster et al. "Counterfactual Multi-Agent Policy Gradients" arXiv v3 2024

---

> > ### Comment · Reviewer_Cnks · 2025-04-02
> >
> > Thank you, Authors, for your rebuttal. Stating that you will include the citations for state-history value functions, you made me more optimistic about this work. However, the most crucial problems of mine with this paper have not been addressed.
> > > 1. I would have to see any preliminary results (possibly under an anonymous link) before I lean positively towards this paper. Crucially, an ablation which compares some (doesn't have to be all) variants of QFix to non-QFix methods, with bigger networks, is a pre-requisite.
> > > 2. It is not true that Definition 3.1 prescribes a unique maximum. Generally, $argmax$ is a set and IGM says that the argmax set of the global value function is the product of argmax sets of local value functions. For example, in case of one-dimensional actions and $N$ agents, if $Q(s,a)=-(a_1 - 1)^2 (a_1 + 1)^2 \dots (a_N - 1)^2 (a_N + 1)^2$, the IGM from Definition 3.1 holds and there are $2^N$ maxima. My original comment remains unchanged.
> > > 3. Regarding Lemma 4.2 and Theorem 4.3 and 4.4: you claim the ease with which your results come to be your strength. If your theoretical results are meant to make a contribution of your paper, the formulated problems shouldn't be too obvious to make proofs about. If something follows too easily, it should be acknowledged. For example, I can now define a function class, let's say $Q(h,a) = b(h) - w(h,a)(u_1\cdot \dots \cdot u_N)^2$ where $w(h,a) > 0$ satisfies your conditions. Of course you can prove the same results about it and it is even simpler than your proposed class. This brings me to another point: words like "minimal" should be used with caution - minimality means something very specific.
> > > 4. Regarding your use of universal function approximation and citation of QPLEX (which I am familiar with): the fact that QPLEX paper does something does not make it correct. QPLEX, just like you in this rebuttal, refers to universal function approximation theorem. But even that theorem (including the version cited by QPLEX (Csaji, 2021)) assumes that the approximated function is continuous. You do not, and I gave you an example when your theorem breaks. Before I consider raising my score, the theoretical limitations of this paper should be addressed, and its rigor improved.
> >
> > **References**
> > Csaji, 2021. Approximation with Artificial Neural Networks.

---

> > > ### Author Response · Authors · 2025-04-08
> > >
> > > We thank the reviewer for the further feedback.
> > > Please see auxiliary figures at [1], which includes:
> > > - Fig 1: Updated results (now 5-6 runs per model per task, also shown as interquartile mean (IQM) [2]), of interest to rev. `5gNj`.
> > > - Fig 2: Probability of improvement [2], of interest to rev. `5gNj`.
> > > - Tab 1, Fig 3: Model size ablation, of interest to revs. `zepP`, `Cnks`.
> > >
> > > 1. We run additional experiments for QMIX-big (QMIX with bigger size) and Q+FIX-mono-small (Q+FIX-mono with smaller size) on all the 5v5 maps. [1, Table 1]. shows all mixer sizes. In terms of size, QMIX-big is comparable to QPLEX and Q+FIX-mono, while Q+FIX-mono-small is comparable to QMIX. [1, Fig 3] contains the ablation results; to avoid clutter, only QMIX, QMIX-big, Q+FIX-mono and Q+FIX-mono-small are shown (other methods are shown in [1, Fig 1]). These results reaffirm that Q+FIX-mono performs well not because of model size, but often in spite of smaller models, and due to our mixing structure. For the final version of the paper, we will extend this ablation to 10v10 and 20v20.
> > > 2. We understand better now that $\argmax$ can itself describe a set of solutions, and does not intrinsically assume a unique maximal element; we agree completely with the reviewer, and will fix the definitions accordingly. We note that this does not affect QIGM or QFIX.
> > > 3.
> > >     - We are unable to concretely understand the reviewer’s concern here; if they are saying that the lemmas/theorems are so obvious they need not be stated or proven, then we strongly disagree, and expect other reviewers would have requested formal proof. If they are saying that they do not meet a threshold of importance to be called theorems, but should, e.g., be reformulated as lesser results like propositions, then that is agreeable and we can make these minor changes. If the concern lies elsewhere, we would appreciate further clarification.
> > >     - The example provided is not just another function class; it is very specifically a special case of QIGM for one specific function $f(u_1, \ldots, u_N) = - \sum_i u_i^2$; This is a perfectly valid special case of QIGM, but our result remains strictly speaking a generalization of any of the provided examples. The importance of proving the more general case is that it is necessary for QFIX. Without the general case over a general class of $f$ functions, we would not be able to define QFIX by having a fixee advantage take the role of $f$. Without proving lemma 4.2 and theorem 4.3 for a general class of functions $f$, QFIX would not exist.
> > >     - We agree that our use of the term "minimal" is not formal and can cause confusion; we will happily replace all mention of "minimality" into something less formal like "simplicity".
> > > 4. We understand the concern better now; universal function approximation theorems (UATs) come in many forms, but not all UATs are exclusively formulated to approximate continuous functions, at least partially because not all refer to the same notion of approximation. The most well known UAT by Cybenko is formulated in terms of uniform convergence, a very strong notion of approximation. However, there are other forms of UAT that use weaker notions of approximation and that are applicable to approximate non-continuous functions. E.g., Hornik’s Theorem [3, Theorem 1] is another popular UAT, and is based on $p$-norm convergence (with $p<\infty$) that proves approximation to $L^p$ functions; this includes large classes of non-continuous functions. In the same document, Hornik informally formulates a corollary that implies another form of approximation for functions that are merely measurable; this is a version of UAT we can employ while making minimal assumptions on $Q$ and $Q_\text{fixee}$.
> > >
> > >     We are happy to clarify these assumptions and conclusions more explicitly; Thm 4.3 is not a statement related to NNs, so it needs no adjustment (it fundamentally states that eqs (38, 39) are the values of $w, b$ sufficient to guarantee QIGM=Q for arbitrary Q). Thm 4.4 does need to be reformulated. We need to assume IGM values that are measurable, and fixees that are also measurable. We must also assume that the fixee’s preimage $A^{-1}_\text{fixee}(0)$ is a measurable set. All of these are fairly mild assumptions. Then, eq (39) is trivially measurable, and eq (38) is measurable as a whole, as it is a piecewise construction based on measurable functions on measurable partitions. Since eqs (38, 39) are measurable, we can apply the corollary informally stated in the discussion section of [3], to justify using neural networks to learn these functions, with the corresponding approximation guarantees on compact subsets of the input space.
> > >
> > > **References**
> > >
> > > 1. https://anonymous.4open.science/r/qfix-icml-rebuttal-5C81/icml-2025-rebuttal.pdf
> > > 2. Agarwal et al., "Deep Reinforcement Learning at the Edge of the Statistical Precipice", NeurIPS 2021.
> > > 3. Hornik, "Approximation Capabilities of Multilayer Feedforward Networks", Neural Networks 4, 1991.

---

### Official Review · Reviewer_5gNj · 2025-03-06

**Overall Recommendation:** 3

**Summary:**

In this paper, the authors address the problem of cooperative multi-agent reinforcement learning with value function decomposition method. They propose a class of decomposition function that are complete with respect to the IGM principles (the max of individual value functions matches the max of the joint value function). The proposed class of function can be applied to previous value function decomposition method an the author show how previous method fit in the proposed theoretical framework.

The theoretical method is used to design several new algorithms that are bringing simplification (in terms of number of parameters) compared to previous work.

The proposed method are evaluated on the SMACv2 multi-agent RL benchmark agains other value decomposition method.

**Update after rebuttal**

Most of my comments were addressed and I am more confident about the results after reading the rebuttal. This paper was interesting and I believe the theoretical discussion is helpful to the field.

**Claims And Evidence:**

The authors claim to "fix" the value decomposition problem. The problem being highlighted as the class of decomposition function used before being either too complex or not expressive enough.

Theorems show that the proposed class of functions encompasses the previous complex functions.

Empirical results show that the proposed algorithm have similar performance to the more complex method (QPLEX).

The claim that the proposed method is more simple is a bit harder to evaluate. The author show that they can devise an value decomposition method with fewer parameters than QPLEX. The derivation of the method does seem a bit simpler than QPLEX but it is quite subjective in my opinion.

A lot of the claim is about "simplicity" and "fixing" the previous work. Although the theoretical approach supports the claim that the proposed algorithm is mathematically sound. The "fixing" is a bit of a stretch since the performance gap does not appear that significant empirically.

**Essential References Not Discussed:**

Not that I could think of.

**Experimental Designs Or Analyses:**

3 runs per model is fairly low, and the “statistical precipice” paper recommends more. Especially given how close the model performance is in Figure 2. In Figure 3 the authors average the performance between 27 runs. It is a bit confusing as of why they don’t directly show it on figure 3.

The authors mention that VDN fails to be a competitive baseline but when one looks at figure 2, VDN is not always the worst, and the asymptotic gap between VDN and the best method seem to be within the confidence interval.

**Methods And Evaluation Criteria:**

The authors evaluate the methods on SMACv2 which is a common MARL benchmark. They use return as opposed to win rate, they give a justification. I am not familiar enough with the benchmark to know if the justification is correct but I don’t think it is unreasonable to look at returns.
The baselines (VDN, QMIX, QPLEX) are relevant. Adding QTRAN would have brought more completeness but it is still ok. It would also have been useful to mention a non value decomposition method, or at least remind the reader whether or not those methods are the state of the art on this benchmark (I remember that MAPPO was quite a strong contender).

**Other Comments Or Suggestions:**

- I think the writing style could be more nuanced when comparing to the related work. The authors make a bold claim of “fixing value decomposition” and highlight the issues in previous work. Although the work of the paper is interesting and impactful, I think the writing could be a bit more modest and consider that the previous methods (even if they have issues) did help in coming up with the new idea. E.g. "QPLEX is more convoluted than our version" => "QPLEX is a specific case of our proposed decomposition scheme."
- Definition 3.3 seems like a lemma or a property rather than a definition?

**Other Strengths And Weaknesses:**

Strengths:

-	The problem formulation and description of the different steps from previous work was very clear and useful to understand the different innovations that happened in value decomposition methods.
-	The theoretical treatment of the value decomposition is comprehensive and provides some harmonization of previous work. I am not sure I would really call it a simplification but at least it gives a sound theoretical framework and it shows well how QPLEX falls into it.

Weaknesses:

-	The proposed theoretical form is helpful in ensuring IGM but it does not guide us to what is the best function class to solve the original problem of collaborative MARL.
-	Figure 2 shows that most of the variant provide the same performance. The theoretical analysis helped in simplifying the model but not in beating QPLEX (although solving some instability issues maybe). I also find it a bit puzzling that the three Q+FIX methods have about the same performance. It would be great if the authors could elaborate on that aspect.

**Questions For Authors:**

- I did not understand the problem mentioned with WQMIX in the related work, could the authors further clarify or provide an example?

- Why do the Q+FIX method have the same performance on the benchmark while the design choices for w and b are different?

**Relation To Broader Scientific Literature:**

They position themselves as an improvement towards previous value decomposition method VDN, QMIX, QPLEX which are correctly presented in the paper. Honorable mentions of other MARL methods could have been useful.

**Theoretical Claims:**

They claim that the proposed value function decomposition scheme is IGM complete and can be used to make previous schemes IGM complete.

I checked Lemma 4.2. I went through theorem 4.3, the proofs make sense to me as one can construct a function Q_IGM from any function in the IGM class.

---

> ### Author Rebuttal · Authors · 2025-04-01
>
> We thank the reviewer for their feedback.
>
> # Direct Questions
>
> ## Re: WQMIX
>
> The WQMIX theory [1] explicitly assumes fully-observable control (MMDP), and makes assumptions that do not hold for Dec-POMDPs, e.g., that decentralized policies can achieve the same optimal behavior as centralized policies.
> We will clarify this in the paper.
>
> ## Re: Similar Performance of Q+FIX variants
>
> We believe the similarity in performance of Q+FIX variants hints at the importance of our mixing:
> - When employing our mixing, it does not matter how "(un)sophisticated" the fixee is; We are able to elevate all fixees equitably. This demonstrates the effectiveness of our structure over model size, and suggests that smaller models may be preferable.
> - When comparing the stability of Q+FIX to the instability of QPLEX, this implies that the complexity/size of QPLEX is a hindrance and that achieving IGM-completeness via simpler models more effective.
>
> We also note that the models of $w, b$ are not different across Q+FIX variants; only the fixee is.
> Perhaps the reviewer is referring to the $w, b$ constructed in the proof of thm 4.3; however, that is a construction to prove IGM-completeness, and not a requirement.
>
> # Other Comments
>
> ## Re: Simplicity
>
> Our claims of simplicity are informed quantitatively (Table 1), and qualitatively (eqs (22,25) are simpler than eq (16)).
> The claims of "fixing" refer to the expansion of the representation class of VDN and QMIX, not performance.
> We will clarify both points in the paper.
>
> ## Re: Statistical Significance
>
> We share the reviewer's concern that modern RL suffers from issues of statistical significance due to the increasing complexity of evaluation.
> While at the time of submission we were limited to 3 seeds per model per task, we have continued to run more evaluations.
> We are now iterating the 5th seed, and the results are consistent, with higher significance.
>
> We note that [3] does not prescribe a high number of seeds, but rather recognizes the practical limitations of evaluation ("3-10 runs are prevalent in deep RL as it is often computationally prohibitive to evaluate more runs"), and makes recommendations that are "easily applicable with 3-10 runs per task".
>
> We already adhere to all applicable recommendations:
> - confidence intervals over point estimates
> - aggregate results over tasks to increase significance (though we use mean, not IQM)
>
> When aggregating using IQM, the distinction between QPLEX and QFIX drops (though QFIX still remains in a slight lead); clearly IQM benefits QPLEX by ignoring its unstable runs. We will include the IQM results in the final paper as well, though we believe it's important to note that QPLEX remains less stable.
>
> It's not clear how to apply other recommendations that require each run to be summarized by a single scalar score; e.g., using final or maximal performances are both unfair, respectively against or in favor of QPLEX.
> We are happy to take suggestions.
>
> ## Re: Fig 3
>
> We thank the reviewer for pointing out an issue in the presentation of fig 3.
> The intent was to aggregate results in accordance to [4], first normalizing returns per task separately (to avoid tasks with wider return ranges from dominating over others), then aggregating across tasks.
>
> We have fixed the plot, and it is almost indistinguishable; this is reasonable, as the return ranges are "similar" (same magnitude).
>
> ## Re: Function Class for MARL
>
> We agree that whether value decomposition methods represent the best function class for coop MARL remains an open question.
> However, that question falls beyond the scope of our work: our goal is to improve upon value decomposition methods themselves.
>
> ## Re: Relationship to QPLEX
>
> Though QPLEX and QFIX are both IGM-complete, QPLEX mixing is not strictly a special case of QFIX mixing: it is not possible to take the QPLEX models $w_i, b_i, \lambda_i$ and construct equivalent QFIX models $w, b$ s.t. QFIX and QPLEX values are equal for all inputs; for equality, the QFIX models must also depend on $Q_i$.
> In that sense, QFIX is not just a reparameterization of QPLEX.
> We will clarify this distinction in the paper.
>
> ## Re: Def 3.3
>
> Defs 3.3, 4.1 are framed in accordance to Def 1 from [5].
> However, we agree and will reframe both.
>
> # Rebuttal Summary
>
> We thank the reviewer for their feedback.
> We believe to have addressed all the stated concerns, and hope they will revisit their evaluation positively.
>
> 1. Tabish et al. "Weighted QMIX: Expanding Monotonic Value Function Factorisation for Deep Multi-Agent Reinforcement Learning" NeurIPS 2020
> 2. Marchesini et al. "On Stateful Value Factorization in Multi-Agent Reinforcement Learning" AAMAS 2025
> 3. Agarwal et al. "Deep Reinforcement Learning at the Edge of the Statistical Precipice" NeurIPS 2021
> 4. Papoudakis et al. "Benchmarking Multi-Agent Deep Reinforcement Learning Algorithms in Cooperative Tasks" NeurIPS 2021
> 5. Wang et al. "QPLEX: Duplex Dueling Multi-Agent Q-Learning" ICLR 2020.

---

> > ### Comment · Reviewer_5gNj · 2025-04-08
> >
> > I appreciate the authors' response to my concerns and updated my score. I think the method is valuable, I am not familiar enough with MoE to judge if the application to MT-MARL is too straightforward as mentioned by Reviewer 8eg2, at least to me it is interesting.

---

> > > ### Author Response · Authors · 2025-04-08
> > >
> > > Firstly, we note that the reviewer’s response to our rebuttal appears to be related to another submission, not ours.
> > >
> > > During the rebuttal period, we were able to update our results in accordance to some concerns raised about statistical significance.
> > > Please see auxiliary figures at [1], which includes:
> > > - Fig 1: Updated results (now 5-6 runs per model per task, also shown as interquartile mean (IQM) [2]), of interest to rev. `5gNj`.
> > > - Fig 2: Probability of improvement [2], of interest to rev. `5gNj`.
> > > - Tab 1, Fig 3: Model size ablation, of interest to revs. `zepP`, `Cnks`.
> > >
> > > We note that the authors of [2] claim that a probability of improvement (POI) that is above 50% with its entire CI indicates a statistically significant result;  out of all methods, Q+FIX-sum is the only one to achieve this against all other methods.
> > >
> > > **References**
> > >
> > > 1. https://anonymous.4open.science/r/qfix-icml-rebuttal-5C81/icml-2025-rebuttal.pdf
> > > 2. Agarwal et al., "Deep Reinforcement Learning at the Edge of the Statistical Precipice", NeurIPS 2021.

---

### Official Review · Reviewer_zepP · 2025-03-13

**Overall Recommendation:** 2

**Summary:**

The paper bridges theory and practice by proposing QFIX , a minimalist yet powerful framework for IGM-complete value decomposition. By extending prior methods with a simple fixing mechanism, QFIX achieves superior performance, stability, and scalability, setting a new standard for cooperative MARL algorithms.

## Update after Rebuttal

The authors have addressed several of my concerns. However, the use of outdated experimental environments and baseline algorithms remains an unresolved issue. While the authors provided some justification for their choices, more recent studies in the MARL community have already adopted modern benchmarks and stronger baselines, making the current experimental setup less compelling. I believe it is important for the community to move away from legacy environments and evaluation protocols to ensure progress and fair comparison. As such, I am keeping my original score. Nevertheless, I believe that updating the experimental section would significantly improve the overall quality and impact of the paper.

**Claims And Evidence:**

This paper is theoretically innovative, with a complete proof process, and the authors have conducted tests on the SMAC benchmark, providing a certain degree of support for their claims.

**Essential References Not Discussed:**

No

**Experimental Designs Or Analyses:**

The experimental benchmarks are reasonably chosen, but the ablation experiments maybe insufficient.

**Methods And Evaluation Criteria:**

The QFIX series of methods and their variants proposed in this paper take into account the issue that value function decomposition in multi-agent reinforcement learning needs to satisfy the IGM property. Their methodological foundation is a minimized IGM formula, which aligns with the requirements of the relevant problems. Meanwhile, the paper adopts SMACv2, a widely recognized benchmark platform in the field of multi-agent reinforcement learning, as the experimental environment, and evaluates performance using metrics such as average return, with win rate results further explained in the appendix. These evaluation criteria effectively reflect the performance of the methods in collaborative tasks.

**Other Comments Or Suggestions:**

No

**Other Strengths And Weaknesses:**

strengths:

1. This article is written in a smooth and coherent manner, maintaining a high level of readability from theoretical derivation to experimental validation. The derivation process of QFIX, Q+FIX, and their variants progresses step by step, starting from the issues with VDN and QMIX, moving on to corrective approaches, and finally introducing an optimized additive correction strategy. This demonstrates strong continuity and coherence in the research line of thought.

2. The method is scalable, as QFIX adopts a "correction" network, which enhances the model's expressive power without altering the core structure of the original approach. It is compatible with VDN and QMIX and can be extended to other non-IGM-complete methods.

Weaknesses:

1. The theoretical proof assumes that the network has sufficient expressive power, but in practical training, the performance is constrained by model capacity and optimization difficulties. The article does not explore the impact of different network architectures and hyperparameters on the final performance.

2. It is unclear whether the performance improvement of the original method by the fixing network is due to the increase in parameters or the enhancement of the method's representational capacity.

**Questions For Authors:**

What is the reason for the improvement in the method's performance due to the correction network? Is it because of the increase in parameters or the enhancement of completeness? This point does not seem to be confirmed by relevant experiments.

**Relation To Broader Scientific Literature:**

The key contributions of the paper are closely related to multiple Literature in the multi-agent reinforcement learning, providing a new perspective for future related research.

**Theoretical Claims:**

I didn't carefully check the proof in this paper, But they seems to be right.

---

> ### Author Rebuttal · Authors · 2025-04-01
>
> We thank the reviewer for their kind words and positive feedback.
>
> # Direct Questions
>
> ## Re: Reason for the Improvement and Model Size
>
> We believe that the empirical results combined with the model sizes of Table 1 provide a compelling argument that the performance of Q+FIX is driven from its mixing mechanism over mixer size, and in almost all cases **despite** smaller mixer size.
> Note that Table 1 shows the sizes of the **mixing** networks alone; this includes the size of the fixee models (when applicable), and excludes the individual value models.
>
> Moreover:
> - VDN uses no mixer (hence no entry in Table 1). Any parameterized mixing method must by definition have more parameters; this is unavoidable.
> - QPLEX employs by far the most parameters than any other method, including all Q+FIX variants.
> - QMIX employs a mixing network of intermediate size (between VDN and QPLEX).  Though Q+FIX-mono employs more parameters than QMIX, this is the only case of a parametric Q+FIX mixer being larger than any other parametric baseline mixer;  again, Q+FIX-{sum,lin} employ significantly fewer parameters than QMIX.
>
> Finally, we note that the performances of all Q+FIX variants are similar, which suggests that **conditioned on the use of our mixing structure**, the quality and size of fixee is not an important factor, and that it is specifically the QFIX mixing structure that drives performance.
> To practitioners, this suggests the use of the smallest models, i.e., Q+FIX-{sum,lin}.
>
> Though we believe the above already provides sufficient evidence for the effectiveness of our mixing structure over model size, we will also run an ablation on model size as requested by reviewer `Cnks`, for the one case where the ablation is feasible, to further prove this point.
>
> # Other Comments
>
> ## Re: Assumption of sufficient expressive power
>
> The assumption of "sufficient expressive power" is an appeal to the Universal Approximation Theorem (UAT) [1, 2], and this methodology is consistent with other works in the literature, most relevant being Proposition 2 from [3].
> Naturally, any concrete architecture may fall short of UAT in practice; this is true of any deep learning model.
> We will clarify this point better in the text.
>
> Re: hyperparameterization: all baselines (VDN, QMIX, QPLEX) employ the parameterization provided by [4], from the corresponding codebase `pymarl2` [5].
> As such, the baseline parameterization has already been optimized by prior work.
> Our mixing models, on the other hand, have not, yet we have found good results even when using the baseline defaults, and without spending significant computational resources towards hyperparameter optimization of Q+FIX.
>
> # Rebuttal Summary
>
> We believe we have addressed the reviewer's concerns, that there are no major concerns remaining, and hope the reviewer will revisit their evaluation accordingly.
> If the reviewer has further concerns, we will gladly use our allotted discussion time to receive further feedback and provide further information.
>
> 1. Cybenko "Approximation by superpositions of a sigmoidal function" MCSS 1989
> 2. Lu et al. "The Expressive Power of Neural Networks: A View from the Width" NeurIPS 2017
> 3. Wang et al. "QPLEX: Duplex Dueling Multi-Agent Q-Learning" ICLR 2020
> 4. Ellis et al. "SMACv2: An Improved Benchmark for Cooperative Multi-Agent Reinforcement Learning" arXiv 2023
> 5. github.com/benellis3/pymarl2

---

> > ### Comment · Reviewer_zepP · 2025-04-02
> >
> > The authors' response has partially addressed some of my concerns, for which I am grateful. However, I still have a few reservations beyond the impact of model size. First, I share the concerns raised by Reviewer Cnks, which have a significant influence on my overall evaluation of the paper.
> >
> > In addition, I am puzzled as to why all the baselines used in the paper are from five years ago. Could QFIX be applied to more recent value decomposition algorithms as well? Moreover, most recent MARL papers accepted at top conferences typically demonstrate the generality of their proposed methods across multiple diverse environments—this is currently lacking in the submission.

---

> > > ### Author Response · Authors · 2025-04-08
> > >
> > > We thank the reviewer for the further feedback.
> > > Please see auxiliary figures at [1], which includes:
> > > - Fig 1: Updated results (now 5-6 runs per model per task, also shown as interquartile mean (IQM) [2]), of interest to rev. `5gNj`.
> > > - Fig 2: Probability of improvement [2], of interest to rev. `5gNj`.
> > > - Tab 1, Fig 3: Model size ablation, of interest to revs. `zepP`, `Cnks`.
> > >
> > > **General Concerns Shared with Reviewer `Cnks`**
> > >
> > > Please see our final response to Reviewer `Cnks`, which addresses all of their remaining concerns. To summarize:
> > > - We ran preliminary ablation results on the sizes of QMIX and Q+FIX-mono, which shows that Q+FIX-mono outperforms QMIX even when QMIX is made bigger and Q+FIX-mono is made smaller. This again confirms that the performance of Q+FIX is driven by the mixing structure.
> > > - The issue with definitions 3.4 and 4.1 are a trivial fix that we will perform, and have no consequence on the rest of the work.
> > > - The claims of IGM-completeness are easily formalized by making mild assumptions and employing other forms of universal function approximation that are applicable to wide sets of measurable functions, not just continuous ones.
> > >
> > > **Can QFIX Be Applied to Other Fixees?**
> > >
> > > QFIX can be applied to any other value function decomposition model; however it only really makes sense to apply it to fixees that are not already IGM-complete.
> > > That is part of why we focus on VDN and QMIX.
> > > Further, our results indicate that QFIX performs best when paired with simpler fixees, as indicated by Q+FIX-sum performing at least marginally better than other variants.
> > >
> > > **More Recent Methods**
> > >
> > > Though newer methods than VDN, QMIX, and QPLEX do exist, to the best of our knowledge, virtually none have withstood the test of time, and VDN, QMIX, and QPLEX still represent the main reference points for value function decomposition methods to this day; e.g.:
> > > - The MARL book from 2024 [3], in the chapter dedicated to value decomposition, only mentions VDN, QMIX, WQMIX, QTRAN, and QPLEX as notable value decomposition methods, i.e., all methods we have discussed in our submission (with a discussion on the limitations of WQMIX and QTRAN in our related work section). FACMAC is also mentioned, but only as a reference to a policy gradient method that employs critic factorization (where IGM is not even a necessary or desired condition).
> > > - The SMACv2 paper from 2023 [4] only uses VDN and QMIX as evaluation baselines belonging to the value function decomposition family.
> > > - The "rethinking implementation details" ICLR blogpost from 2023 [5] focuses exclusively on QMIX.
> > >
> > > These are all primary modern up-to-date reference points for value function decomposition, and they hardly focus on anything more than VDN, QMIX, QPLEX (and sometimes WQMIX and QTRAN).
> > >
> > > **Evaluation Environments**
> > >
> > > Even though other evaluation environments do exist in the literature, SMAC variations (SMACv1, SMACv2, SMAX, etc) remain by far the most popular environment for cooperative MARL in the literature (not least because it has high variability in terms of maps, setups, team sizes, procedurally generated scenarios, etc).
> > >
> > > While other environments do exist in the literature, we disagree that an evaluation encompassing a significantly wider variety than ours is a common standard.
> > > Among others, we can point to the following seminal and primal works:
> > > - Both QMIX papers [6, 7] (both conference and journal versions) perform evaluations exclusively on matrix games (of little interest to us) and SMAC scenarios.
> > > - WQMIX [8] performs evaluations primarily on SMAC scenarios (though they do include one small evaluation on Predator Prey, this is a small component of their evaluation).
> > > - QPLEX [9] performs evaluations exclusively on SMAC scenarios.
> > > - The "rethinking implementation details" ICLR blogpost [5] performs evaluations exclusively on SMAC scenarios.
> > >
> > > All of these are seminal works in top-tier conferences, and their methods were almost exclusively evaluated on SMAC.
> > >
> > > **References**
> > >
> > > 1. https://anonymous.4open.science/r/qfix-icml-rebuttal-5C81/icml-2025-rebuttal.pdf
> > > 2. Agarwal et al., "Deep Reinforcement Learning at the Edge of the Statistical Precipice", NeurIPS 2021.
> > > 3. Albrecht et al., "Multi-Agent Reinforcement Learning: Foundations and Modern Approaches". MIT Press, 2024.
> > > 4. Ellis et al. "SMACv2: An Improved Benchmark for Cooperative Multi-Agent Reinforcement Learning" NeurIPS (Datasets and Benchmarks Track) 2023.
> > > 5. https://iclr-blogposts.github.io/2023/blog/2023/riit/
> > > 6. Rashid et al. "QMIX: Monotonic Value Function Factorisation for Deep Multi-Agent Reinforcement Learning" ICML 2018.
> > > 7. Rashid et al. "Monotonic Value Function Factorisation for Deep Multi-Agent Reinforcement Learning" JMLR 2020.
> > > 8. Rashid et al. "Weighted QMIX: Expanding Monotonic Value Function Factorisation for Deep Multi-Agent Reinforcement Learning" NeurIPS, 2020.
> > > 9. Wang et al. "QPLEX: Duplex Dueling Multi-Agent Q-Learning" ICLR 2020.

---

### Decision · Program_Chairs · 2025-05-01

**Decision:**

Reject

**Comment:**

This work presents 3 variants of QFIX. They are supported by intuitions and some derivations. Experiments (post-rebuttal) seem promising.

However, the rebuttal is too heavy and the updates needed are too much, for the manuscript to be accepted at its current stage. I suggest the authors incorporate the experiments and also rewrite the derivation part to avoid any theoretical overclaim.

Some parts of the reviews believe that the environment tasks and the baselines are outdated. Some reviewers and I do not see them as outdated and I therefore did not take this part of the reviews into consideration.